# SVL: Empowering Spiking Neural Networks for Efficient 3D Open-World Understanding

Xuerui Qiu [1 2 3]  Shaowei Gu [1 2 3]  Peixi Wu [4]  JiaKui Hu [5]  Yaozhi Wen [1 2 3]  Yuqi Pan [1]  Xinhao Luo [1]  Bo XU [1]
Guoqi Li [1]

## Abstract

Spiking Neural Networks (SNNs) offer an energy–efficient route to 3D spatio–temporal perception, yet they lag behind Artificial Neural Networks (ANNs) due to weak pretraining and heavy inference stacks, limiting generalization and multimodal reasoning (e.g., zero–shot 3D classification and open–world QA). We present a universal **S**pike–based **V**ision–**L**anguage pretraining framework (SVL) that equips SNNs with open–world 3D understanding while preserving end–to–end spike efficiency. SVL comprises two core components: (i) Multi–scale Triple Alignment (MTA), a label–free triplet contrastive objective aligning 3D, image, and text; and (ii) Re–parameterizable Vision–Language Integration (Rep–VLI), which converts offline text embeddings into lightweight weights for text–encoder–free inference. Moreover, we present the first fully spike–driven point Transformer, Spike-driven PointFormer, whose 3D spike–driven self–attention (3D-SDSA) reduces interactions to sparse additions, enabling faster, more efficient training. Extensive experiments show that SVL attains strong zero–shot 3D classification (85.4% top–1) and consistently outperforms prior SNNs on downstream tasks (e.g., +6.1% 3D cls, +2.1% DVS actions, +1.1% detection, +2.1% segmentation) while enabling open–world 3D question answering, sometimes outperforming ANNs. To the best of our knowledge, SVL represents the first scalable, generalizable, and hardware-friendly paradigm for 3D open-world understanding, effectively bridging the gap between SNNs and ANNs in complex open-world understanding tasks. Code is available at SVL.

## 1. Introduction

Bio-inspired Spiking Neural Networks (SNNs) offer an efficient approach to learning superior representations from sparse 3D geometric data (e.g., event streams and point clouds) (Roy et al., 2019), owing to their distinctive spike-driven nature (Pei et al., 2019) and spatio-temporal processing capabilities (Maass, 1997). For instance, the Speck (Yao et al., 2024) chip uses event-by-event sparse processing to handle 3D input data, with operational power consumption as low as 0.7 mW. However, existing SNNs (Qiu et al., 2025a; Yao et al., 2025; Zhou et al., 2024) exhibit a significant performance gap compared to ANNs, and remain task-dependent, lacking both generalizable representations and the ability to achieve multimodal understanding in 3D open-world scenarios.

For instance, when deploying SNNs in real-world scenarios (Yao et al., 2024), they may struggle to generalize to input data from unseen categories not present in the training set. This highlights the critical need to develop robust pretraining strategies to enhance the visual representation capabilities and adaptability of SNNs. Existing methods, such as STDP-based initialization (Lee et al., 2018) and knowledge distillation in SpikeBert and SpikeCLIP (Lv et al., 2023; Bal & Sengupta, 2024; Lv et al., 2025), refine spike-based representations, while SpikformerV2 and Spike-driven Transformer V3 (Yao et al., 2025; Zhou et al., 2024) employ masked image modeling to improve scalability. However, these approaches (Lee et al., 2018) either lose effectiveness as dataset complexity increases, demand substantial computational resources (Zhou et al., 2024), which limits their feasibility for neuromorphic hardware deployment, or lack multimodal integration (Lv et al., 2023). Moreover, pre-trained models often exhibit inadequate visual representation capabilities and limited transferability, restricting unified applicability to downstream tasks (Yao et al., 2025).

Another challenge is the limited availability of annotated 3D datasets, as the creation of such datasets is both labor-

[1]Institute of Automation, Chinese Academy of Sciences [2]School of Future Technology, University of Chinese Academy of Sciences [3]Zhongguancun Academy [4]University of Science and Technology of China [5]Peking University. Correspondence to: Guoqi Li <guoqi.li@ia.ac.cn>.

*Proceedings of the 43rd International Conference on Machine Learning*, Seoul, South Korea. PMLR 306, 2026. Copyright 2026 by the author(s).

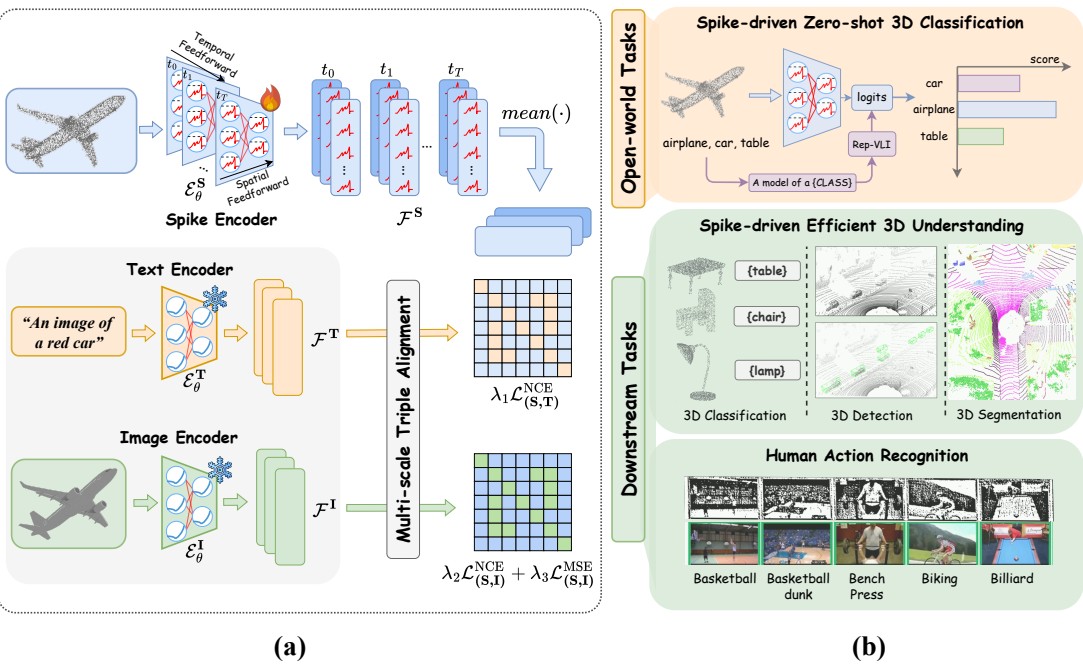

**(a)**             **(b)**

*Figure 1.* Overall architecture and applications of our SVL. (a) In pretraining, we proposed Multi-scale Triple Alignment (MTA) that jointly optimizes correlation alignment across text, image, and 3D inputs. (b) For downstream tasks, we propose Re-parameterizable Vision-Language Integration (Rep-VLI) to reparameterize the text embeddings generated by the text encoder into lightweight weights, enabling efficient spike-driven inference.

intensive and error-prone, rendering it often impractical for large-scale, real-world applications (Xu et al., 2024; Liu et al., 2025). In response, Vision-Language Models (VLMs) (Radford et al., 2021; Miao et al., 2025; Tan et al., 2025) have been employed to explore the transfer of knowledge gleaned from extensive 2D datasets to facilitate open-world 3D understanding. However, most VLMs (Xue et al., 2023; 2024; Liu et al., 2023b) depend heavily on large-scale text encoders during inference, which imposes substantial limitations on the practicality of hardware deployment.

In this paper, we introduce a universal **S**pike-based **V**ision-**L**anguage pretraining framework (SVL) that enhances SNNs' capability for open-world multimodal 3D understanding while maintaining efficient spike-driven inference. As shown in Fig. 1, SVL incorporates two key innovations: (i) Multi-scale Triple Alignment (MTA), which enables label-free triplet representation learning to capture the geometric properties of 3D data, and (ii) Re-parameterizable Vision–Language Integration (Rep–VLI), which converts offline text embeddings into lightweight weights for text–encoder–free deployment. We leverage CLIP for its strong generalization: during pretraining, CLIP is frozen and a spike-driven 3D encoder is aligned to CLIP's image/text spaces via contrastive learning; the pre-trained 3D model is then fine-tuned for downstream tasks. In addition to SVL , we present the first fully spike–driven

point Transformer, Spike-driven PointFormer, whose 3D spike–driven self–attention (3D-SDSA) reduces interactions to sparse additions, enabling faster, more efficient training that supports large–scale pretraining and generalizes broadly across 3D tasks. Our main contribution can be summarized as:

- We propose two key innovations about SVL: (i) Multi-scale Triple Alignment (MTA), a label-free triplet learning mechanism for capturing geometric properties of 3D data across different scales, and (ii) Re-parameterizable Vision-language Integration (Rep-VLI), which achieves lightweight inference by reducing the computational overhead of text encoder.

- We present the first fully spike–driven point Transformer. Its 3D spike–driven self–attention (3D-SDSA) reduces interactions to sparse additions, enabling faster, more efficient training that supports large–scale pretraining and generalizes broadly across 3D tasks.

- Extensive experiments on multiple benchmarks demonstrate the effectiveness of SVL, achieving state-of-the-art (SOTA) performance in both 3D open-world understanding and downstream tasks, as well as generative applications such as 3D object captioning and open-world question answering.

## 2. Related Works

**Pretraining Algorithms of SNNs**   Numerous pretraining methods are proposed for spike-based representation learning. (Lee et al., 2018) utilized spike-timing-dependent plasticity (Bliss & Collingridge, 1993) to initialize SNNs, enhancing the model's robustness and training speed. While this approach has been successful on simple datasets with shallow networks, its effectiveness diminishes as the complexity of the datasets and networks increases. To address these issues, SpikeBert and SpikeCLIP (Lv et al., 2023; Bal & Sengupta, 2024; Lv et al., 2025) employ a two-stage knowledge distillation process from ANNs to enhance spike-based representations for complex downstream tasks. However, these methods rely on ANN weight initialization, limiting structural flexibility. Additionally, they use LayerNorm, which hinders neuromorphic hardware deployment. SpikformerV2 and Spike-driven Transformer V3 (Yao et al., 2025; Zhou et al., 2024; 2023) apply a masked image modeling approach to address the performance degradation in SNNs as the model scales up. However, they require substantial storage and computational resources, and the lack of multimodal integration, particularly language guidance, limits their effectiveness in open-world understanding tasks.

**Vision-language Models (VLMs)**   aim to align image and text embeddings for cross-modal transfer, with CLIP (Radford et al., 2021; Zhao et al., 2025; Chen et al., 2025; Li et al., 2021a) being a seminal work that uses contrastive learning for zero-shot classification. Building on this foundation, subsequent methods have expanded cross-modal alignment to include other modalities. These approaches typically fall into two categories: dual-encoder and triple-encoder frameworks. Dual-encoder fine-tune both visual and textual encoders (Lv et al., 2025; Zhang et al., 2021; Zhu et al., 2022). Triple-encoder frameworks incorporate additional modality-specific encoders (Xue et al., 2023; 2024; Zeng et al., 2023; Su et al., 2025), which combine triple models to achieve open-world understanding. This architecture is highly flexible, making it suitable for a variety of downstream tasks (Xu et al., 2024; Liu et al., 2023b; Wu et al., 2025a; Li et al., 2025). However, triple-encoder frameworks still rely on large text encoders during inference, hindering hardware deployment.

**Efficient 3D recognition**   from sparse, irregular data (events, point clouds) follows two main directions: point-based pipelines that operate on raw points to extract geometric features (Qi et al., 2017; Wang et al., 2019), and voxel-based pipelines that discretize into regular grids and apply sparse 3D convolutions (Wu et al., 2015). While deeper voxel/backbone designs can improve accuracy, the gains often come with substantial compute and memory costs, hindering deployment. To reduce cost, the SNN community

has integrated spiking neurons with point-based models for low-power edge computing (Ren et al., 2024; Wu et al., 2024; Zhou et al., 2025); early designs, however, tend to be task-specific and capacity-limited. E-3DSNN (Qiu et al., 2025a) advances this line with spike sparse convolutions, delivering strong results across multiple 3D tasks while preserving spike-driven operation. Orthogonal to encoder choice, SVL pretraining enhances representation quality and enables open-world multimodal understanding while retaining spike efficiency.

Transformer-style spiking architectures are emerging. Spike Point Transformer (Wu et al., 2025b) introduces a Transformer-based SNN but still uses non-spiking operators and applies temporal encoding to point clouds, which degrades energy efficiency and slows training. Spike PointNet (Zhou et al., 2025) and E-3DSNN (Qiu et al., 2025a) rely on point-wise or sparse-convolutional inductive biases that may limit expressiveness and scalability. In contrast, we propose Spike PointFormer, the first fully spike-driven point Transformer: its 3D spike-driven self-attention (SDSA) performs addition-only interactions on spike tensors, enabling energy-efficient large-scale pretraining and broad generalization, and serving as a complementary architecture to SVL pretraining.

## 3. Preliminaries

**Spiking Neurons**   are inspired by the dynamics of biological neurons (Maass, 1997; Li et al., 2023), which are the fundamental units of Spiking Neural Networks (SNNs). Among these, the Leaky Integrate-and-Fire (LIF) neuron is the most widely used due to its balance between biological plausibility and computational efficiency (Maass, 1997). We begin by translating the LIF spiking neuron into an iterative expression using the Euler method (Wu et al., 2018), which is described as follows:

$$u_i^{(\ell)}[t+1] = h_i^{(\ell)}[t] + f(w^{(\ell)}, x_i^{(\ell-1)}[t]), \qquad (1)$$

$$s_i^{(\ell)}[t] = \Theta(u_i^{(\ell)}[t+1] - \vartheta), \qquad (2)$$

$$h_i^{(\ell)}[t+1] = \beta u_i^{(\ell)}[t+1](1 - s_i^{(\ell)}[t]), \qquad (3)$$

Here, $\beta$ is the time constant $t$ and $i$ represents the time step and the neuron index in the $\ell$-th layer, respectively. The weight matrix $w$ defines the synaptic connections between adjacent layers, while $f(\cdot)$ is a function that denotes operations such as convolution (Conv) or fully connected (FC). The input is represented by $x$, and $\Theta(\cdot)$ denotes the Heaviside step function. When the membrane potential $u$ exceeds the firing threshold $\vartheta$, the LIF neuron generates a spike, $s$. Additionally, $h$ represents the membrane potential after the spike event, which is scaled by a constant factor $\beta$.

Directly training the above LIF-based SNNs requires the use of backpropagation through time (BPTT) (Wu et al.,

2018), resulting in a time complexity of $\mathcal{O}(LT)$, where $L$ and $T$ are the number of layers and time steps. This significantly increases both the training time and memory requirements. To mitigate this issue, we use the Integer LIF Spiking Neuron.

**Integer LIF Spiking Neuron** is incorporated into our SVL to reduce the quantization error, training time, and memory (Yao et al., 2025; Luo et al., 2024; Qiu et al., 2025b), which allows us to rewrite Eq. (2) as:

$$s_i^{(\ell)}[t] = \lfloor \text{clip}\{u^{(\ell)}[t], 0, D^t\} \rceil, \tag{4}$$

where $\lfloor \cdot \rceil$ denotes the rounding operator, $\text{clip}\{x, a, b\}$ confines $x$ within range $[a, b]$, and $D^t$ is a hyperparameter indicating the maximum emitted integer value by I-LIF. Moreover, I-LIF will emit integer values while pretraining and convert them into binary spikes by expanding the virtual timestep to ensure that the inference is spike-driven with only sparse addition.

## 4. Method

Our primary goal is to develop a spike-based encoder that accurately captures the geometric properties of 3D input data and efficiently achieves a unified representation for open-world 3D understanding with the spike-driven nature. To this end, we construct our triplet dataset $\{(D_1^t, I_1^t, T_1^t), (D_2^t, I_2, T_2^t), \cdots, (D_n^t, I_n^t, T_n^t)\}$, which consists of a 3D input $D_i^t$, an image $I_i$, and a text description $T_i$ at $t$ time step.

### 4.1. 3D Input Representation

In this part, we present the 3D input representation, such as point clouds and event streams. Event streams, in particular, require special handling. We define them as $E_i = (x_i, y_i, t_i, p_i)$. Using a sliding window technique (Wang et al., 2019; Ren et al., 2024), we convert event streams into an event cloud, formulated as:

$$E_i = (x_i, y_i, z_i) \quad \text{where} \quad z_i = \frac{t_i - t_{\min}}{t_{\max} - t_{\min}}, \tag{5}$$

By doing so, we treat event streams as a distinct kind of spatio-temporal point cloud. This allows us to consider both point clouds and event streams as collections of points, denoted by $D^t = \{\mathcal{P}, \mathcal{F}\}$. This includes voxel sets $D_k^t = \{\mathcal{P}_k^t, \mathcal{F}_k^t\}$, where $\mathcal{P}_k^t \in \mathbb{R}^3$ represents the 3D coordinates and $\mathcal{F}_k^t \in \mathbb{R}^D$ indicate the features across $d$ channels at the time step $t$. Following this, we utilize our I-LIF spiking neuron to encode these 3D inputs into spatio-temporal spike trains, which are then transmitted to the spike encoder.

### 4.2. Multi-scale Triple Alignment

To develop a unified representation for open-world 3D understanding, we introduce a multi-scale triple alignment (MTA) framework that jointly optimizes correlation alignment across text, image, and 3D inputs. This framework integrates both semantic spike-text alignment and fine-grained spike-image alignment. Specifically, the overall architecture of SVL, illustrated in Fig. 1, comprises three encoders: (i) Text Encoder ($\mathcal{E}_\theta^T$): embeds text into text features $\mathcal{F}^T \in \mathbb{R}^C$; (ii) Spike-based Encoder ($\mathcal{E}_\theta^S$): transforms spike inputs into spike trains $\mathcal{F}^S \in \mathbb{R}^{T \times C}$. (iii) Image Encoder ($\mathcal{E}_\theta^I$): encodes images into image features $\mathcal{F}^I \in \mathbb{R}^C$. Here, $C$ represents the embedding dimension. These encoders collaboratively embed the triplet texts, spikes, and images into their respective feature spaces, facilitating comprehensive and fine-grained alignment across different modalities.

**Semantic Spike-Text Alignment** To leverage the open-world recognition capabilities of the pre-trained CLIP model (Radford et al., 2021), we align the spike firing rate $\mathcal{F}^S/T$ with the text embeddings $\mathcal{F}^T$ obtained from CLIP, using a spike-text tuple $\mathcal{B}_i = \{T_i^t, \mathcal{D}_i^t\}$ as input. The core idea is to bring the feature centroids of 3D instances and their corresponding text prompts closer together in the embedding space. To achieve this, we compute the InfoNCE loss (van den Oord et al., 2018) between the mean spike trains and the text features, as follows:

$$\mathcal{L}_{(\mathbf{S},\mathbf{T})}^{\text{NCE}} = -\frac{1}{2|\mathcal{B}|} \sum_i^{|\mathcal{B}|} \log \frac{e^{\tau \mathbf{x}_i \cdot \mathbf{y}_i}}{\sum_j^{|\mathcal{B}|} e^{\tau \mathbf{x}_i \cdot \mathbf{y}_j}} + \log \frac{e^{\tau \mathbf{x}_i \cdot \mathbf{y}_i}}{\sum_j^{|\mathcal{B}|} e^{\tau \mathbf{x}_j \cdot y_i}}, \tag{6}$$

where $\mathbf{x}_i = \frac{\mathcal{F}^S/T}{||\mathcal{F}^S/T||_2}$ and $\mathbf{y}_i = \frac{\mathcal{F}^T}{||\mathcal{F}^T||_2}$ represent the normalized spike and text features, respectively. The indices $i$ and $j$ are used for sampling, the dot product ($\cdot$) denotes cosine similarity between vectors, and $\tau$ is a learnable temperature parameter.

**Fine-grained Spike-Image Alignment** A singular spike-text alignment fails to fully capture the semantic information embedded within both images and 3D data. To achieve a more comprehensive multimodal understanding, we further introduce an alignment between image and spike features. Specifically, we first employ the InfoNCE loss to align the image features, denoted as $\mathcal{F}^I$, with the average pulse signals, represented as $\mathcal{F}^S/T$. This alignment can be expressed as follows:

$$\mathcal{L}_{(\mathbf{S},\mathbf{I})}^{\text{NCE}} = -\frac{1}{2|\mathcal{C}|} \sum_i^{|\mathcal{C}|} \log \frac{e^{\tau \mathbf{a}_i \cdot \mathbf{b}_i}}{\sum_j^{|\mathcal{C}|} e^{\tau \mathbf{a}_i \cdot \mathbf{b}_j}} + \log \frac{e^{\tau \mathbf{a}_i \cdot \mathbf{b}_i}}{\sum_j^{|\mathcal{B}|} e^{\tau \mathbf{a}_j \cdot b_i}}, \tag{7}$$

where $\mathcal{C}_i = \{I_i^t, \mathcal{D}_i^t\}$ a spike-image tuple, $\mathbf{a}_i = \frac{\mathcal{F}^S/T}{||\mathcal{F}^S/T||_2}$ and $\mathbf{b}_i = \frac{\mathcal{F}^I}{||\mathcal{F}^I||_2}$ represent the normalized spike and text

features, respectively. However, this approach resulted in overly coarse alignment granularity, failing to account for the fine-grained and tightly coupled alignment between spikes and images. To address this, we incorporate the MSE loss on the basis of the InfoNCE loss to enhance the alignment granularity. The alignment objective between spike trains and images, which is formulated as follows:

$$\mathcal{L}_{(\mathbf{S},\mathbf{I})}^{\text{MSE}} = \sum_i^{|\mathcal{C}|} ||\mathcal{F}_i^{\mathbf{S}} - \mathcal{F}_i^{\mathbf{I}}||^2, \qquad (8)$$

where $|| \cdot ||^2$ is the $\ell_2$ norm. Finally, we obtain the resultant total learning objective $\mathcal{L}_{\text{total}}$ as the combination of $\mathcal{L}_{(\mathbf{S},\mathbf{T})}$ and $\mathcal{L}_{(\mathbf{S},\mathbf{I})}$, where both alignments of semantic spike-text and fine-grained spike-image alignment are injected as:

$$\mathcal{L}_{\text{total}} = \lambda_1 \mathcal{L}_{(\mathbf{S},\mathbf{T})}^{\text{NCE}} + \lambda_2 \mathcal{L}_{(\mathbf{S},\mathbf{I})}^{\text{NCE}} + \lambda_3 \mathcal{L}_{(\mathbf{S},\mathbf{I})}^{\text{MSE}}, \qquad (9)$$

where $\lambda_1$, $\lambda_2$, and $\lambda_3$ are hyperparameters that balance the influence of image features, text features, and spike trains. For our main experiments, we set all three to 1, with a detailed ablation study on their impact presented in Tab 6.

### 4.3. Re-parameterizable Vision-Language Integration

While pretrained vision-language models excel at zero-shot transfer, their inference phase typically requires a large, computationally expensive text encoder. This presents a significant bottleneck for SNNs, undermining their inherent efficiency and complicating deployment on neuromorphic hardware.

To overcome this, we introduce the Re-parameterizable Vision-Language Integration (Rep-VLI) module. Rep-VLI's core innovation is to pre-compute and embed textual information directly into the weights of a lightweight classification layer, completely discarding the text encoder during inference.

Specifically, for a set of $K$ candidate text prompts $\{T_1, T_2, \ldots, T_K\}$, we use the text encoder $\mathcal{E}_\theta^T$ to generate a corresponding weight matrix $W^L \in \mathbb{R}^{K \times C}$:

$$W_j^L = \tau \mathcal{E}_\theta^T(T_j), \qquad (10)$$

where $W_j^L$ is the re-parameterized weight vector for the $j$-th text prompt.

During inference, we adopt a hardware-friendly spike-count decision rule instead of a conventional softmax. For an input 3D data point $D_i$, our spike-based encoder $\mathcal{E}_\theta^S$ produces spike features $\mathbf{s}[t] \in \{0,1\}^C$ over $T$ timesteps. The predicted class, $\hat{y}_i$, is the one whose corresponding weight vector aligns best with the accumulated spike activity:

$$\text{logits}_i = \arg\max_j \frac{1}{T} \sum_{t=1}^{T} W_j^L \cdot \mathcal{E}_\theta^S(D_i^t), \qquad (11)$$

This process selects the class with the highest firing rate, preserving a fully spike-driven and hardware-compatible inference path. Ultimately, Rep-VLI elegantly sidesteps the need for a persistent text encoder at inference time. By re-parameterizing textual knowledge into a compact weight matrix, it ensures our model remains lightweight and perfectly aligned with the operational principles of spike-based neuromorphic systems.

### 4.4. Spike-driven 3D Encoder

**Backbone Suite** We instantiate SVL with three spike-driven 3D backbones: (i) Spike PointNet (Wu et al., 2024) for lightweight point-cloud processing; (ii) E-3DSNN (Qiu et al., 2025a), a sparse spike-convolutional backbone suited to edge deployment; and (iii) Spike-driven PointFormer, our fully spike-driven Transformer for high-capacity cloud settings.

**Spike-driven PointFormer** As shown in Fig. 3, given a point set $\mathcal{P}_k \in \mathbb{R}^{T \times N \times 3}$, we form local neighborhoods by farthest-point sampling and $k$-NN grouping:

$$X = \text{KNN}(\text{FPS}(P)), \qquad X \in \mathbb{R}^{T \times N' \times 3}, \qquad (12)$$

A learnable add-only pointwise embedding (addition only since inputs are spikes), followed by an I-LIF neuron $\mathcal{SN}(\cdot)$, produces spike features:

$$S = \mathcal{SN}(\text{MLP}(X)), \qquad S \in \mathbb{R}^{T \times N' \times D}, \qquad (13)$$

We then perform local-to-global feature extraction with $L$ Spike-driven PointFormer layers (SDF):

$$\begin{aligned} f_0 &= \text{EMP}(S), \\ f_\ell &= \text{SDF}(f_{\ell-1}) + f_{\ell-1}, \quad \ell = 1, \ldots, L, \end{aligned} \qquad (14)$$

where EMP denotes element-wise max pooling over the neighborhood axis and all $f_\ell \in \mathbb{R}^{T \times N' \times D}$.

Spike-driven self-attention inside SDf. From $f_{\ell-1}$, three add-only linear maps yield $Q, K, V \in \mathbb{R}^{T \times N' \times D}$, which are converted to spikes:

$$Q_S = \mathcal{SN}(Q), \quad K_S = \mathcal{SN}(K), \quad V_S = \mathcal{SN}(V), \qquad (15)$$

A matrix-multiplication variant (3D-SDSA) used in Spike-driven PointFormer is

$$\begin{aligned} \text{SDSA}_{\text{3D}}(Q_S, K_S, V_S) &= \mathcal{SN}(Q_S(K_S^\top V_S)) \\ &= \mathcal{SN}((Q_S K_S^\top)V_S), \end{aligned} \qquad (16)$$

Here all products are spike-driven matmuls that reduce to sparse additions via address-event accumulation algorithms (Horowitz, 2014), preserving end-to-end spike computation.

*Table 1.* 3D Zero-shot classification results on the large-scale Objaverse-LVIS (Obj.) (Deitke et al., 2023) and ModelNet40 (M40.) (Wu et al., 2015) datasets. "*" denotes self-implementation results with open-source code. "Energy" denotes the estimated energy consumption, following (Qiu et al., 2025a; Yao et al., 2023); further details are provided in Appendix C. "Point+Text" denotes the parameters of the point encoder and the text encoder.

| Architecture | Model | Pre-train Method | Input | $T \times D$ | Point+Text Param (M) | Energy (mJ) | Obj. | M40. |
|---|---|---|---|---|---|---|---|---|
| | PointCLIP (Zhang et al., 2021) | N/A | Image | N/A | 25.5+57.3 | 24.9+20.3 | 1.9 | 20.2 |
| ANN | PointNet (Qi et al., 2017) | Openshape (Liu et al., 2023b) | Point | N/A | 3.47+202.5 | 20.1+71.7 | 24.4 | 74.9 |
| | Point-Bert (Yu et al., 2022) | | Point | N/A | 21.9+202.5 | 83.2+71.7 | 43.2 | 82.8 |
| | Sparseconv (Graham et al., 2017) | | Voxel | N/A | 5.3+202.5 | 0.61+71.7 | 31.7 | 78.8 |
| | | | Voxel | N/A | 41.3+202.5 | 2.13+71.7 | 43.4 | 83.4 |
| | Point-Bert (Yu et al., 2022) | Ulip (Xue et al., 2023) | Point | N/A | 21.9+227.8 | 81.2+80.6 | 34.9 | 69.6 |
| | Point-Bert (Yu et al., 2022) | Ulip2 (Xue et al., 2024) | Point | N/A | 21.9+202.5 | 83.7+71.7 | 50.6 | 84.7 |
| SNN | SpikeCLIP* (Lv et al., 2025) | N/A | Image | 4×1 | 9.5+22.8 | 10.6+0.41 | 0.5 | 5.1 |
| | Spike PointNet (Ren et al., 2024) | SVL (Ours) | Point | 1×4 | 3.57 | 0.27 | 24.9 | 76.3 |
| | **Spike-driven PointFormer-S (Ours)** | | Point | 1×4 | 7.69 | 5.1 | 40.1 | 82.1 |
| | **Spike-driven PointFormer-L (Ours)** | | Point | 1×4 | 22.1 | 9.4 | 43.4 | 83.1 |
| | E-3DSNN-T (Qiu et al., 2025a) | | Voxel | 1×4 | **2.10** | **0.04** | 33.6 | 79.6 |
| | E-3DSNN-S (Qiu et al., 2025a) | | Voxel | 1×4 | 3.51 | 0.09 | 36.4 | 81.3 |
| | E-3DSNN-L (Qiu et al., 2025a) | | Voxel | 1×4 | 17.7 | 0.64 | 43.9 | 84.6 |
| | E-3DSNN-H (Qiu et al., 2025a) | | Voxel | 1×4 | 46.7 | 0.79 | **47.0** | **85.4** |

*Table 2.* 3D object captioning results on Objaverse-LVIS. "*" indicates SVL-13B is prompted for shorter captions with no more than 20 words. The evaluation utilizes a range of metrics, including Sentence-BERT, SimCSE, BLEU-1, ROUGE-L, and METEOR.

| Method | Vision Encoder | LLM | Input | S.-BERT | SimCSE | B-1. | R-L. | MET. |
|---|---|---|---|---|---|---|---|---|
| InstructBLIP-13B (Dai et al., 2023) | ViT (Dosovitskiy, 2020) | Vicua | Image | 45.90 | 48.86 | 4.65 | 8.85 | 13.23 |
| LLaVA-13B (Liu et al., 2023a) | | | Image | 46.37 | 45.90 | 4.02 | 8.15 | 12.58 |
| PointLLM-13B (Xu et al., 2024) | PointBert (Yu et al., 2022) | Vicua | Point | 47.91 | 49.12 | 3.83 | 7.23 | 12.26 |
| PointLLM-13B* (Xu et al., 2024) | PointBert (Yu et al., 2022) | | Point | 50.15 | **50.83** | 17.09 | 20.99 | 16.45 |
| **SVL-13B (Ours)** | Spike-driven PointFormer-L | | Point | 44.87 | 45.91 | 3.77 | 6.85 | 12.25 |
| **SVL-13B (Ours)*** | Spike-driven PointFormer-L | | Point | 47.80 | 47.08 | 11.45 | 14.69 | 16.40 |
| **SVL-13B (Ours)*** | Spike-driven PointFormer-L | SpikeLLM (Xing et al., 2025) | Point | **51.21** | 50.18 | **18.45** | **21.32** | **18.40** |
| Human | N/A | N/A | N/A | 100.00 | 100.00 | 100.00 | 100.00 | 100.00 |

# 5. Experiments

To validate that our spike-based 3D encoder learns robust visual representations via SVL, we evaluate it on diverse 3D open-world tasks, including zero-shot classification and visual question answering. The pretrained encoder is also fine-tunable for downstream tasks like 3D classification, segmentation, detection, and action recognition. This section details the experimental setup, including backbones, datasets, and implementation, followed by quantitative results and ablation studies on modality count, time steps, and loss functions. For additional speed and accuracy comparisons between our Spike-driven PointFormer and other spike-based 3D encoders—as well as temporal scene understanding results and CLIP encoder sizes—see **Appendix A**. Implementation details and dataset descriptions are provided in **Appendix D** and **Appendix F**, respectively.

## 5.1. 3D Open-world Understanding

**Zero-shot classification** We evaluate the zero-shot classification performance of our models on the widely-used ModelNet40 (Wu et al., 2015) and the larger, more challenging Objaverse-LVIS (Deitke et al., 2023). Compared to other benchmarks, Objaverse-LVIS offers broader class coverage and a long-tailed distribution, providing a more realistic evaluation of open-world 3D understanding (Liu et al., 2023b). As shown in Tab. 1, our SVL-based E-3DSNN achieves 85.4% accuracy on ModelNet40 with only 17.7M parameters, outperforming both ANN and SNN baselines. This demonstrates SVL's effectiveness in enhancing both accuracy and efficiency.

Specifically, compared to OpenShape and ULIP, our model achieves 85.4% accuracy (vs. 83.4% and 69.6%), consumes only 0.79 mJ of energy (vs. 161.8 mJ and 73.8 mJ), and uses fewer parameters (17.7M vs. 41.3M and 21.9M). It also delivers a 15.8% accuracy gain over ULIP-based Point-BERT (Xue et al., 2023) while consuming just 11.4% of the

*Table 3.* 3D Downstream Tasks: 3D classification results on ModelNet40 (M-40) (Wu et al., 2015) and ScanObjectNN (Scan-O) (Uy et al., 2019).

| Architecture | Model | Input | Param (M) | Energy (mJ) | $T \times D$ | ModelNet40 | ScanObjectNN |
|---|---|---|---|---|---|---|---|
| ANN | PointNet (Qi et al., 2017) | Point | 3.27 | 2.02 | N/A | 89.2 | 68.2 |
| | PointNet + ULIP (Xue et al., 2023) | Point | 3.47 | 2.34 | N/A | 92.1 | 72.1 |
| | Pointformer (Zhao et al., 2021) | Point | 4.91 | 30.1 | N/A | 92.8 | 81.3 |
| SNN | Spike Point TransFormer (Wu et al., 2025b) | Point | 9.6 | 21.1 | $4 \times 1$ | 88.5 | 80.1 |
| | P2SResLNet (Wu et al., 2024) | Point | 14.3 | - | $4 \times 1$ | 88.7 | 81.2 |
| | SpikingPointNet (Lan et al., 2023) | Point | 3.47 | 0.91 | $16 \times 1$ | 88.6 | 66.6 |
| | Spike PointNet (Ren et al., 2024) | Point | 3.47 | 0.24 | $1 \times 4$ | 88.2 | 70.0 |
| | Spike PointNet + SVL | Point | 3.47 | 0.27 | $1 \times 4$ | **90.1** (↑ 1.9) | **76.1** (↑ 6.1) |
| | E-3DSNN-S (Qiu et al., 2025a) | Voxel | 3.27 | 0.02 | $1 \times 4$ | 91.7 | 78.7 |
| | E-3DSNN-S + SVL | Voxel | 3.27 | 0.02 | $1 \times 4$ | **92.7** (↑ 1.0) | **80.9** (↑ 2.2) |
| | E-3DSNN-L (Qiu et al., 2025a) | Voxel | 17.7 | 0.26 | $1 \times 4$ | 91.2 | 80.2 |
| | E-3DSNN-L + SVL | Voxel | 17.7 | 0.31 | $1 \times 4$ | **93.7** (↑ 2.5) | **83.0** (↑ 2.8) |
| | **Spike-driven PointFormer-S (Ours)** | Point | 7.69 | 5.1 | $1 \times 4$ | 92.6 | 82.1 |
| | **Spike-driven PointFormer-L (Ours)** | Point | 22.1 | 9.4 | $1 \times 4$ | 92.1 | 81.7 |
| | **Spike-driven PointFormer-L (ours) +SVL** | Point | 22.1 | 9.8 | $1 \times 4$ | **93.9** (↑ 1.8) | **83.4** (↑ 1.7) |

*Table 4.* 3D Downstream Tasks: 3D segmentation, detection, and human action recognition results on KITTI (Geiger et al., 2012a), Semantic KITTI (Behley et al., 2019), DVS Action (Miao et al., 2019), and DVS128 Gesture (Amir et al., 2017). Moreover, for the DVS dataset, we adopt a pre-training timestep of $1 \times 4$, consistent with other datasets, and a fine-tuning timestep of $6 \times 4$.

| Architecture | Method | $T \times D$ | KITTI AP-E (%) | Semantic KITTI mIoU (%) | DVS Action Acc. (%) | DVS128 Gesture Acc. (%) |
|---|---|---|---|---|---|---|
| ANN | E-3DANN (Qiu et al., 2025a) | N/A | 89.4 | 69.4 | - | - |
| | PointNet (Qi et al., 2017) | N/A | - | 14.6 | 75.1 | 95.3 |
| SNN | E-3DSNN (Qiu et al., 2025a) | $1 \times 4$ | 89.6 | 68.5 | - | - |
| | E-3DSNN + SVL | $1 \times 4$ | **90.7** (↑ 1.1) | **69.7** (↑ 1.2) | - | - |
| | Spike PointNet (Ren et al., 2024) | $1 \times 4 / 6 \times 4$ | - | 12.1 | 78.4 | 96.9 |
| | Spike PointNet + SVL | $1 \times 4 / 6 \times 4$ | - | **15.6** (↑ 2.1) | **80.5** (↑ 2.1) | **98.5** (↑ 1.6) |

energy. On Objaverse-LVIS, our model performs comparably to ULIP-2 (Xue et al., 2024) but with a 204× energy efficiency advantage. This is enabled by our Rep-VLI module, which reparameterizes text embeddings into compact, spike-compatible weights for zero-shot inference, preserving the spike-driven nature of the encoder. Compared to prior SNN approaches such as SpikeCLIP (Lv et al., 2025), SVL substantially improves the visual representation capacity of spike-based encoders in zero-shot 3D tasks.

**Generative 3D Object Captioning and Open-world Question Answering** We combine the SVL-trained Spike PointFormer with a language model via the LLaVA framework (Liu et al., 2023a) for multimodal pre-training and fine-tuning (see Appendix H). On the 3D object captioning benchmark, prompted with "Describe this 3D model in detail," our SVL-13B achieves performance comparable to state-of-the-art ANN methods (Tab. 2). Semantic metrics (Sentence-BERT, SimCSE) confirm strong alignment

with human references. Notably, as the first SNN-based method in 3D captioning, SVL-13B achieves comparable annotation quality compared to PointLLM. We also evaluate on 3D question answering. As shown in Fig. 2, the model effectively interprets shape, material, function, and context, including visual and functional cues, while demonstrating commonsense reasoning. Despite lacking dense textures, SVL-13B achieves strong perception-language alignment, comparable to ANN models across diverse object types.

### 5.2. 3D Downstream Tasks

**3D Classification, Segmentation, and Detection** We first fine-tuned our models on 3D classification datasets such as ModelNet40 (Wu et al., 2015) and ScanObjectNN (Uy et al., 2019) to evaluate the 3D visual representation capabilities acquired through SVL pretraining. As shown in Tab. 5, the SVL pretraining significantly enhances performance, with the E-3DSNN (Qiu et al., 2025a) and the Spike PointNet

([Ren et al., 2024](#)) architecture achieving improvements of 1.9% and 1.0%, respectively, on ModelNet40. On the more challenging ScanObjectNN dataset, the Spike PointNet architecture demonstrates a substantial accuracy increase, rising from 70.0% to 76.1%. Subsequently, we extended our fine-tuning experiments to datasets such as Semantic KITTI ([Behley et al., 2019](#)) and KITTI ([Geiger et al., 2012a](#)). As illustrated in Tab. 2, our SVL pretraining delivers marked improvements in both 3D segmentation and detection tasks, with the E-3DSNN ([Qiu et al., 2025a](#)) exhibiting performance gains of 1.1% and 1.2%, respectively.

**Human Action Recognition** We further fine-tune our SVL-pretrained spike-based encoder on DVS datasets, including DVS128 Gesture ([Amir et al., 2017](#)) and DVS Action ([Miao et al., 2019](#)), to assess spatiotemporal feature extraction. During pretraining, the I-LIF time step was set to 1 for efficiency, then increased to 6 during evaluation to better capture temporal dynamics. We adopt the point-based method from ([Wang et al., 2019](#)) for efficient DVS data processing (see Section 4.1). As shown in Tab. 4, SVL-pretrained E-3DSNN and Spike Point improve by 2.1% and 1.6% on DVS Action and DVS128 Gesture, respectively, indicating strong scalability and temporal modeling ability of SVL-trained SNNs.

**Comparisons between different Backbone** Tab. 5 compares ANN- and SNN-based 3D backbones on ModelNet40 and ScanObjectNN, reporting input modality, parameter count, measured energy, and the temporal configuration $T \times D$. Among ANN models, Pointformer attains strong accuracy (92.8% on M-40 / 81.3% on Scan-O), while KPConv remains competitive on Scan-O (85.3%). Voxelized efficient baselines (E-3DANN-S) offer favorable energy profiles but slightly lower accuracy on Scan-O.

Within SNNs, prior point-based methods (e.g., P2SResLNet, SpikingPointNet, Spike PointNet) reduce energy but lag in accuracy, and voxel SNNs (E-3DSNN-S,L) strike a different tradeoff with very low energy but moderate robustness on Scan-O.

Our **Spike-driven PointFormer** closes the accuracy gap to leading ANN point backbones while retaining SNN efficiency. The 7.69M-parameter variant achieves **92.6%** on M-40 and **82.1%** on Scan-O at **5.1 mJ** with a $1 \times 4$ temporal setup, surpassing Pointformer on Scan-O with substantially lower energy. The larger 22.1M variant yields similar accuracy (92.1% / 81.7%) at 9.4 mJ. These results indicate that spike-driven transformers can deliver ANN-level recognition performance on point clouds while maintaining the energy advantages characteristic of SNNs.

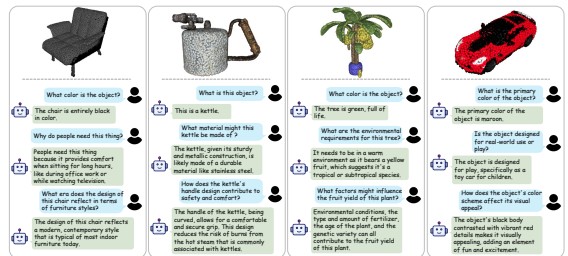

*Figure 2.* Dialogues between SVL-13B and a human user. The dialogues show SVL's ability to understand point clouds' shapes, appearances, functionalities, etc. Additionally, SVL demonstrates abilities to respond to human instructions with common sense, avoiding biases.

### 5.3. Ablation Study

**The Effectiveness of Our MTA** An ablation study was conducted to examine the impact of different loss function combinations during our multi-scale triple alignment (MTA). Specifically, we compared performance with and without the semantic spike-text alignment ($e.g., \mathcal{L}_{(\mathbf{S},\mathbf{T})}^{\text{NCE}}$) and fine-grained spike-image alignment ($e.g., \mathcal{L}_{(\mathbf{S},\mathbf{I})}^{\text{NCE}}, \mathcal{L}_{(\mathbf{S},\mathbf{I})}^{\text{MSE}}$). As shown in Tab. 6, the ablation study highlights the importance of combining loss functions for optimal performance. In the absence of any loss functions, the model only gets 0.5% accuracy on the large-scale Objaverse-LVIS and 5.1% on ModelNet40. Introducing spike-image alignment yields a significant improvement, while the inclusion of semantic spike-text alignment alone demonstrates limited effectiveness. The highest performance is attained when all three loss functions, including the MSE-based fine-grained alignment, are employed, achieving 33.6% accuracy on the large Objaverse-LVIS ([Deitke et al., 2023](#)) and 79.6% on ModelNet40 ([Wu et al., 2015](#)). These findings underscore the synergistic relationship between semantic and fine-grained alignment in enhancing the model's representational capabilities, showing the effectiveness of our MTA module.

**Different Time Steps and Firing Bits** We systematically study time steps ($T$) and firing bits ($D$) for SVL pretraining and downstream fine-tuning (Tab. 7). During pretraining, enlarging $T$ offers negligible accuracy gains yet degrades efficiency: for example, with $D$=1, increasing $T$ from 2 to 4 improves Objaverse-LVIS zero-shot accuracy by only 0.2% while roughly doubling power and worsening latency. In contrast, scaling $D$ at a fixed $T$ consistently improves accuracy and can even reduce power by concentrating information into fewer, stronger spikes. For fine-tuning, higher $T$ can help recognition quality, but the benefit comes with longer inference and higher energy. In practice, we recommend small $T$ (often $T$=1) with moderate $D$ for pretraining to minimize compute, then using modest $T$ (e.g., 2–4) and tuning $D$ for downstream tasks to balance accuracy against latency and power.

*Table 5.* 3D Downstream Tasks: 3D classification results on ModelNet40 (M-40) (Wu et al., 2015) and ScanObjectNN (Scan-O) (Uy et al., 2019).

| Architecture | Model | Input | Param (M) | Energy (mJ) | $T \times D$ | ModelNet40 | ScanObjectNN |
|---|---|---|---|---|---|---|---|
| ANN | PointNet (Qi et al., 2017) | Point | 3.27 | 2.02 | N/A | 89.2 | 68.2 |
| | Pointformer (Zhao et al., 2021) | Point | 4.91 | 30.1 | N/A | 92.8 | 81.3 |
| | Spike Point TransFormer (Wu et al., 2025b) | Point | 9.6 | 21.1 | $4 \times 1$ | 88.5 | 80.1 |
| | KPConv (Thomas et al., 2019) | Point | 14.3 | - | N/A | 92.9 | 85.3 |
| | 3DShapeNets (Wu et al., 2015) | Voxel | 6.97 | 0.61 | N/A | 88.2 | - |
| | E-3DANN-S (Qiu et al., 2025a) | Voxel | 3.27 | 0.13 | $1 \times 4$ | 91.7 | 79.7 |
| SNN | P2SResLNet (Wu et al., 2024) | Point | 14.3 | - | $4 \times 1$ | 88.7 | 81.2 |
| | SpikingPointNet (Lan et al., 2023) | Point | 3.47 | 0.91 | $16 \times 1$ | 88.6 | 66.6 |
| | Spike PointNet (Ren et al., 2024) | Point | 3.47 | 0.24 | $1 \times 4$ | 88.2 | 70.0 |
| | E-3DSNN-S (Qiu et al., 2025a) | Voxel | 3.27 | 0.02 | $1 \times 4$ | 91.7 | 78.7 |
| | E-3DSNN-L (Qiu et al., 2025a) | Voxel | 17.7 | 0.26 | $1 \times 4$ | 91.2 | 80.2 |
| | **Spike-driven PointFormer (Ours)** | Point | 7.69 | 5.1 | $1 \times 4$ | 92.6 | 82.1 |
| | **Spike-driven PointFormer (Ours)** | Point | 22.1 | 9.4 | $1 \times 4$ | 92.1 | 81.7 |

*Table 6.* Ablation study of MTA.

| $\mathcal{L}^{\text{NCE}}_{(\mathbf{S},\mathbf{T})}$ | $\mathcal{L}^{\text{NCE}}_{(\mathbf{S},\mathbf{I})}$ | $\mathcal{L}^{\text{MSE}}_{(\mathbf{S},\mathbf{I})}$ | Obj. | M40. |
|---|---|---|---|---|
| ✗ | ✗ | ✗ | 0.5 | 5.1 |
| ✗ | ✓ | ✗ | 24.8 | 73.1 |
| ✓ | ✗ | ✗ | 21.9 | 70.1 |
| ✓ | ✓ | ✗ | 31.7 | 77.8 |
| ✓ | ✓ | ✓ | **33.6** | **79.6** |

*Table 7.* Ablation study of the pretrain timesteps.

| Method | $T \times D$ | Power (mJ) | Obj. (%) | M40. (%) |
|---|---|---|---|---|
| ANN* | N/A | 0.13 | 34.1 | 81.3 |
| SNN | $1 \times 2$ | **0.02** | 32.9 | 78.5 |
| | $2 \times 1$ | 0.03 | 32.7 | 78.0 |
| | $2 \times 2$ | 0.08 | **33.9** | **80.5** |
| | $1 \times 4$ | 0.04 | 33.6 | 79.6 |
| | $4 \times 1$ | 0.10 | 32.9 | 78.6 |

## 6. Conclusion

In this work, we introduce SVL, a novel spike-based vision–language pretraining framework designed to equip Spiking Neural Networks with robust open-world 3D understanding capabilities while simultaneously preserving their inherent energy efficiency. Through the strategic integration of Multi-scale Triple Alignment and a Reparameterizable Vision–Language Integration module, our approach effectively bridges the longstanding gap between the low-power advantages characteristic of SNNs and the strong generalization capabilities typically associated with advanced vision–language models. Our comprehensive evaluations, which span diverse tasks including zero-shot 3D classification, semantic segmentation, and human action recognition, demonstrate that SVL consistently outperforms prior SNN-based approaches. Furthermore, it achieves performance levels that rival state-of-the-art Artificial Neural Networks, all while operating at a significantly lower computational cost. Notably, SVL empowers SNNs with the unprecedented ability to perform open-world 3D question answering, which marks a significant milestone in the field of multimodal representation learning for spike-based systems. Finally, we present the Spike-driven PointFormer, which stands as the first fully spike-driven point Transformer. This architecture features a 3D spike-driven self-attention mechanism that reduces complex interactions to sparse additions, thereby delivering faster and more energy-efficient training dynamics while maintaining strong predictive accuracy.

## Acknowledgements

This work was partially supported by the Zhongguan-cun Academy (Grant No.s 1800302002), National Distinguished Young Scholars (62325603), National Natural Science Foundation of China (62236009, U22A20103, U2541222), CAS Project for Young Scientists in Basic Research (YSBR116), Beijing Science and 698 Technology Plan (Z241100004224011), and CAAITencent Rhino-Bird Open Research Fund.

## Impact Statement

This paper advances the development of SNNs for open–world 3D understanding, bridging the gap between SNNs and ANNs while preserving energy efficiency. There are many potential societal consequences of our work, none of which we feel must be specifically highlighted here.

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

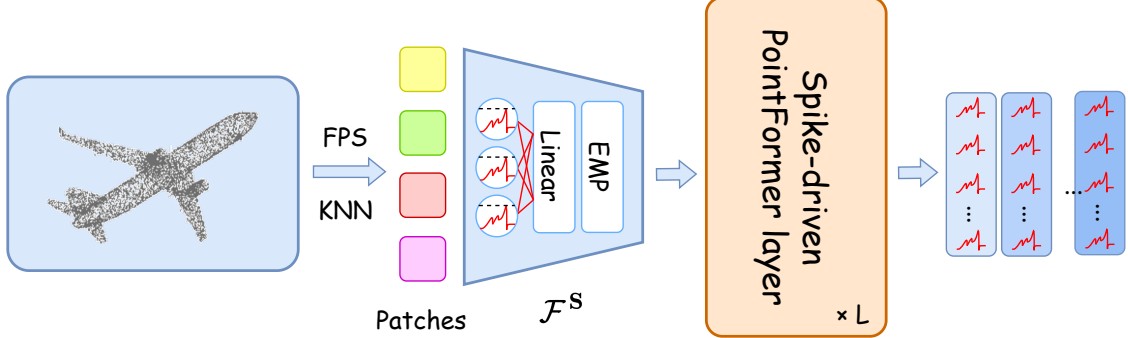

*Figure 3.* Overall architecture of our Spike-driven PointFormer.

# A. More Experiments

This section augments the main results with three complementary studies: comparisons between different Backbone, dynamic scene segmentation on Synthia 4D (Ros et al., 2016) to assess temporal generalization, ablations on event/discretization design (voxel size and point density), and the impact of CLIP image-encoder size used to build vision–language prototypes. For each study, we report results and provide a concise analysis.

### A.1. Speedup across spike point Transformers

We compare training/inference efficiency of our Spike-driven PointFormer against prior spike-based point Transformers (Wu et al., 2025b) and an ANN backbone under the same data loader, batch size, and GPU settings (details in Appendix D). Our design uses a shallow-by-time but deeper-by-layer configuration ($T{\times}D{=}1{\times}4$) with 3D-SDSA, whereas prior spike models adopt $4{\times}1$. As shown in Tab. 8, **PointFormer-S** reaches **100 ms** train / **56 ms** infer with **3.7 GB / 2.5 GB** memory, delivering up to **4.3$\times$** faster training and **4.1$\times$** lower training memory than Spike Point Transformer-1024 (431 ms / 15.2 GB), while also surpassing the ANN Point Transformer-L in both runtime and memory. The larger **PointFormer-L** remains efficient (159 ms / 82 ms; 5.5 GB / 3.5 GB), achieving **2.7$\times$** faster training and **2.8$\times$** lower training memory than the 1024-d spike baseline. These results validate that shifting temporal depth into learnable layers and replacing heavy interactions with sparse additions yields substantial speed/memory benefits without sacrificing accuracy.

*Table 8.* Ablation study of different backbone efficiency on ModelNet40.

| Methods | $T \times D$ | Training | | Inference | |
|---|---|---|---|---|---|
| | | Runtime | Memory | Runtime | Memory |
| Point Transformer-L (Zhao et al., 2021) | N/A | 150ms | 5.1G | 79ms | 3.2G |
| Spike Point Transformer-512 (Wu et al., 2025b) | 4×1 | 326ms | 9.7G | 191ms | 5.2G |
| Spike Point Transformer-768 (Wu et al., 2025b) | 4×1 | 385ms | 12.5G | 201ms | 7.3G |
| Spike Point Transformer-1024 (Wu et al., 2025b) | 4×1 | 431ms | 15.2G | 227ms | 9.5G |
| **Spike-driven PointFormer-S (Ours)** | 1×4 | **100ms** | **3.7G** | **56ms** | **2.5G** |
| **Spike-driven PointFormer-L (Ours)** | 1×4 | **159ms** | **5.5G** | **82ms** | **3.5G** |

### A.2. Dynamic Scene Segmentation on Synthia 4D

We evaluate temporal scene understanding on Synthia 4D using voxelized inputs and multi-step spike simulation. SVL improves the SNN backbone and matches or surpasses ANN counterparts under comparable capacity.

SVL yields a substantial gain over the SNN backbone without pretraining (80.05 vs. 76.41 mIoU) and slightly surpasses a capacity-matched ANN (79.54 mIoU), indicating that aligning spike features to the vision–language space improves scene-level generalization; moreover, increasing temporal extent from $1 \times 2$ to $3 \times 2$ frames further boosts performance (78.91 to 80.05 mIoU), suggesting that spike-driven temporal accumulation is effectively exploited.

*Table 9.* Scene segmentation on Synthia 4D (Ros et al., 2016). SVL enables strong temporal understanding for SNNs.

| Method | Input | Frames | Params (M) | mIoU (%) |
|---|---|---|---|---|
| 3D MinkNet14 (Ros et al., 2016) | Voxel | 1 | 19.3 | 76.24 |
| 4D MinkNet14 (Ros et al., 2016) | Voxel | 3 | 23.7 | 77.46 |
| Same-structure ANN | Voxel | 3 | 19.1 | 79.54 |
| E-3DSNN-L (w/o SVL) | Voxel | $3 \times 2$ | 19.1 | 76.41 |
| E-3DSNN-L (SVL, ours) | Voxel | $1 \times 2$ | 17.7 | 78.91 |
| E-3DSNN-L (SVL, ours) | Voxel | $3 \times 2$ | 19.1 | **80.05** |

## A.3. Event Construction Ablations

We ablate voxel size and the number of input points on ModelNet40 zero-shot with E-3DSNN-T + SVL.

*Table 10.* Ablations on discretization. Moderate voxel size (0.02) and sufficient point density (10k–20k) work best.

| Voxel size | 0.01 | 0.02 | 0.04 |
|---|---|---|---|
| Top-1 Acc. (%) | 78.9 | **79.6** | 79.1 |

| #Points | 5k | 10k | 20k |
|---|---|---|---|
| Top-1 Acc. (%) | 77.3 | 79.6 | **79.8** |

Accuracy peaks at a moderate voxel size of 0.02, where overly fine discretization (0.01) fragments geometry and increases sparsity noise while overly coarse discretization (0.04) washes out structure; likewise, raising point density from 5k to 10k yields a clear improvement whereas the gain from 10k to 20k is marginal, indicating diminishing returns and suggesting that a mid-range density strikes the best balance between fidelity and efficiency.

## A.4. CLIP Encoder Size

We study the impact of the image encoder size used to build vision–language prototypes during pretraining.

*Table 11.* Larger vision encoders yield stronger zero-shot 3D alignment.

| Image encoder | ModelNet40 Top-1 (%) | Objaverse-LVIS Top-1 (%) |
|---|---|---|
| OpenCLIP ViT-B | 66.8 | 25.1 |
| OpenCLIP ViT-G | 70.6 | 27.9 |
| OpenCLIP ViT-bigG | **79.6** | **33.6** |

Scaling the vision encoder substantially strengthens zero-shot transfer (e.g., ViT-B to bigG: +12.8 on ModelNet40 and +8.5 on Objaverse-LVIS), reflecting richer visual semantics that better anchor the spike encoder in the shared space; notably, because Rep-VLI removes the text encoder at inference, larger CLIP backbones increase precomputation and training cost but do not affect runtime latency or energy, making bigG preferable when resources allow.

## B. Open-world Multimodel Learning Details

In this section, we present the details of how to perform open-world multimodal learning after pretraining with the SVL model. Our primary goal is to effectively leverage the capabilities of the pretrained LLM and the spike-based encoder pretrained with SVL. The network architecture is shown in Fig. 4. We select vicuna as our LLM $\mathcal{F}_\theta(\cdot)$, parameterized by $\theta$, use the Spike-driven Pointformer $\mathcal{E}_\theta^S(\cdot)$ pretrained with SVL as the spiking visual encoder, and the projector $\mathcal{P}_\theta(\cdot)$.

For the input 3D data $D_k^t$, we first utilize the pretrained spike-based encoder $\mathcal{E}_\theta^S(\cdot)$ to provide 3D spatiotemporal visual features. Subsequently, a simple linear layer $\mathcal{P}_\theta(\cdot)$ is employed to connect the 3D spatiotemporal visual features to the word embedding space. Specifically:

$$V^t = \mathcal{P}_\theta(\mathcal{E}_\theta^S(D_k^t)), \tag{17}$$

where $V^t$ is the vison tokens at the $t$ time step. Then we concatenate it with the text tokens $I^t$ obtained after tokenization and send the combined features to the LLM $\mathcal{F}_\theta(\cdot)$.

$$R^t = \mathcal{F}_\theta(I^t; V^t), \tag{18}$$

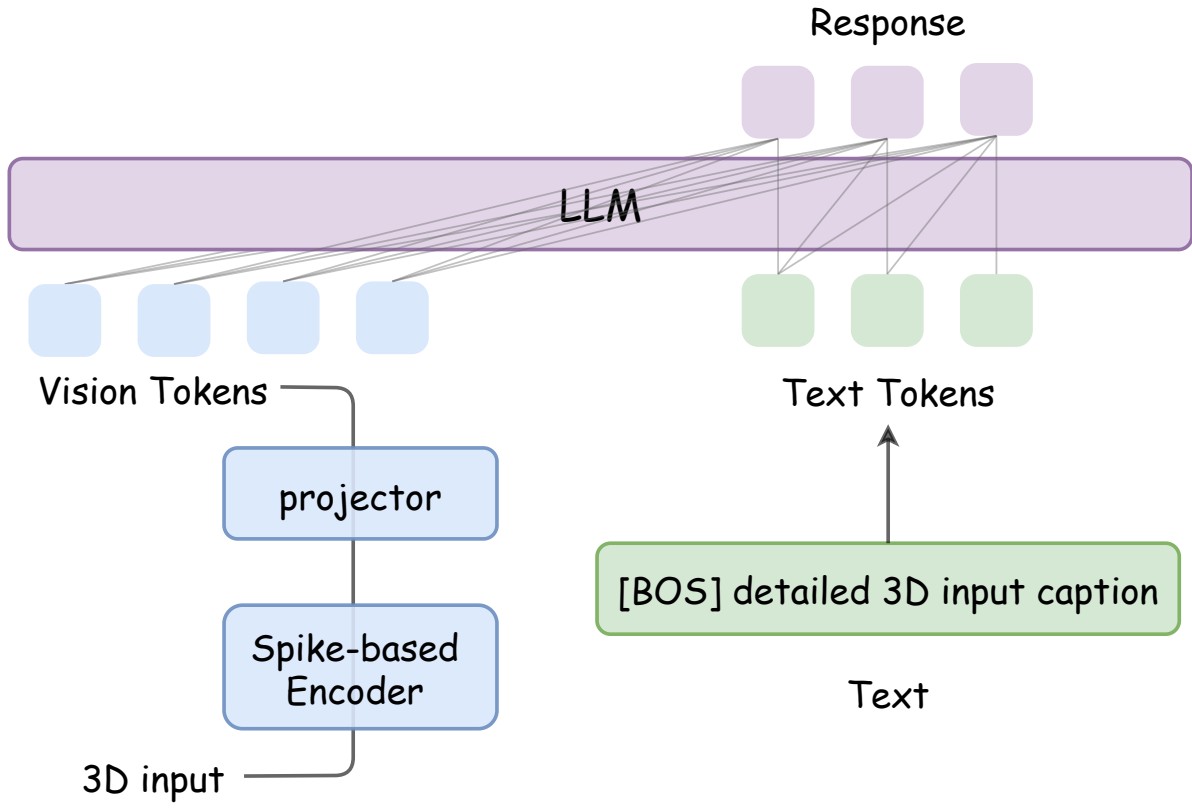

*Figure 4.* Architecture details of open-world multimodel learning.

where $R^t$ is the output response or logits. Our training is divided into two stages. In the first stage, we train the projector while freezing the LLM and the spike-based encoder. In the second stage, we train the LLM and the projector.

*Table 12.* **Qualitative comparisons.** We show the qualitative results of models on the ScanNet (Dai et al., 2017). Our SVL-13B can understand 3D semantics and respond to prompts effectively comparable to other ANN-based models.

| **SVL-13B (Ours)** | (The outputs for ScanNet-Scene0024_02 are shown below.) |
|---|---|
| 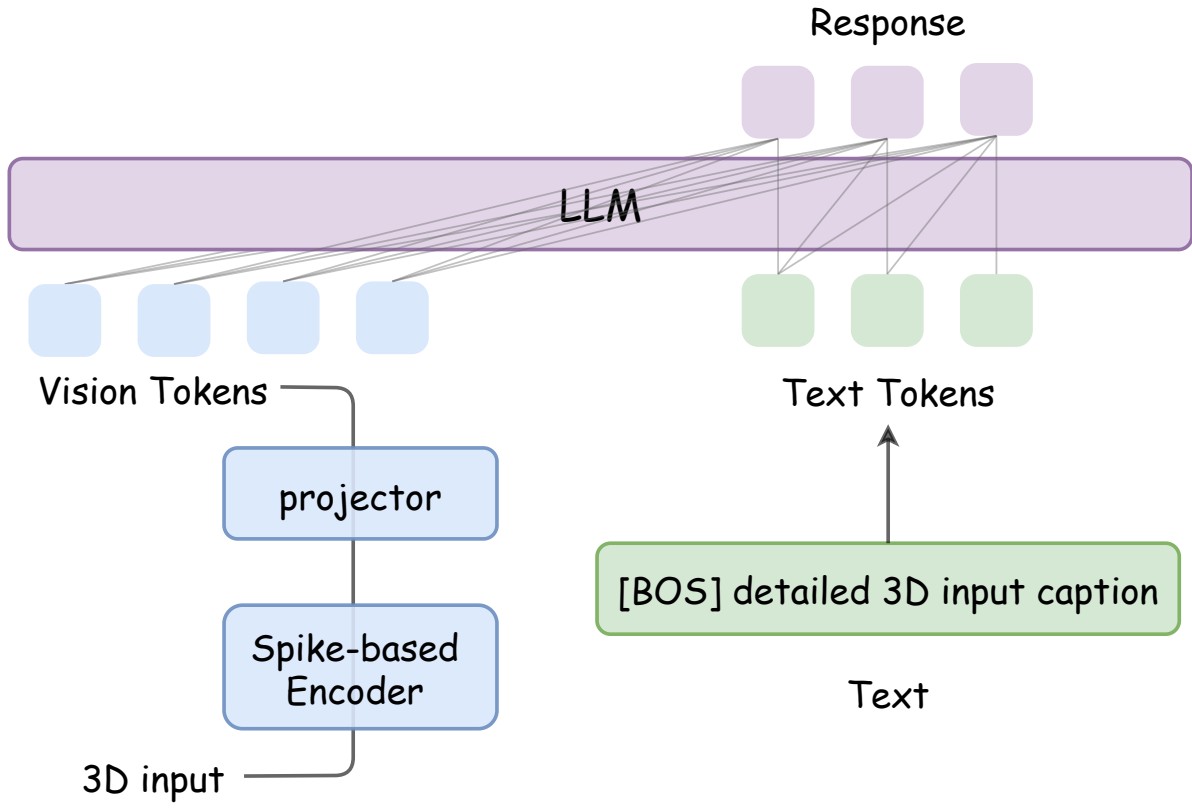 | **This 3D model depicts a traditional-style house with a tasteful aesthetic. The house features a rich brown color on its exterior walls, giving it a warm and welcoming appearance. It has a distinctive architectural design, with a slanted roof that is typical of traditional homes. The interior reflects a beautiful contrast with lighter-colored walls, providing a homely and comfortable ambiance. This model can be used in architectural designing, virtual reality games, or various design projects.** |

## C. Theoretical Energy Consumption

In our SVL framework, Rep-VLI can transform the text embeddings into tiny weights during inference. Additionally, the framework can convert matrix multiplication into sparse addition, which can be implemented as addressable additions on

*Table 13.* **Qualitative comparisons.** We show the qualitative results of models on ModelNet40 (Wu et al., 2015) and Objaverse (Deitke et al., 2023). Our SVL-13B can understand 3D semantics and respond to prompts effectively comparable to other ANN-based models.

| Samples |  |  |
|---|---|---|
| Ground Truth | Laptop | A cartoon black monster like a dragon |
| Prompt | What is this? | Briefly caption this 3D model. |
| InstructBLIP (Dai et al., 2023) | symbol letter l | a black lizard with a sharp tooth in a dark room |
| LLaVA (Liu et al., 2023a) | A small, grainy, black and white letter j. | A 3D model of a dark, menacing dragon. |
| 3D-LLM (Hong et al., 2023) | - | A black and white tiger with long legs, standing on its hind leg. |
| Point-Bind LLM (Guo et al., 2023) | This is a laptop computer. | The 3D model features a large, ornate gargoyle with a horned helmet, sitting on top of a building. |
| PointLLM (Xu et al., 2024) | The 3D model represents a notebook computer, typically a laptop. | The 3D model depicts a menacing black dragon, with its mouth opened wide revealing a row of sharp teeth. |
| **SVL-13B (Ours)** | **This is a 3D model of a laptop.** | **This is a 3D model of a toy dinosaur, which stands upright on its hind legs. It has a spiked back, reflecting its distinctive defense mechanism.** |

neuromorphic chips. In the first coding layer, convolution operations act as Multiply-Accumulate (MAC) operations that convert analog inputs into spikes, similar to direct coding-based SNNs (Wu et al., 2019). Similarly, in the final layer, logit calculations also perform MAC operations. In contrast, in the SNN architecture, the convolution (Conv) or fully connected (FC) layers transmit spikes and execute Accumulation (AC) operations to accumulate weights for postsynaptic neurons. Hence, the inference energy cost for our SVL framework can be expressed as follows:

$$E_{total} = E_{MAC} \cdot (FL_{conv}^1 + FL_{conv}^{VLI}) + E_{AC} \cdot T \sum_{n=2}^{N} FL_{conv}^n \cdot fr^n, \tag{19}$$

where $N$ and $M$ represent the total number of sparse spike convolutions, and $E_{MAC}$ and $E_{AC}$ are the energy costs associated with MAC and AC operations, respectively. The variables $fr^m$, $fr^n$, $FL_{conv}^n$, and $FL_{fc}^m$ denote the firing rate and FLOPs of the $n$-th sparse spike convolution layer. Previous SNN studies (Horowitz, 2014; Rathi & Roy, 2021; Qiu et al., 2024; 2023) assume a 32-bit floating-point implementation in 45nm technology, with $E_{MAC} = 4.6$ pJ and $E_{AC} = 0.9$ pJ for various operations.

Additionally, batch normalization (BN) operations can be fused into the convolutional layers, further reducing computation overhead. Since Rep-VLI eliminates the text encoder during inference, layer normalization (LN) layers are also unnecessary, simplifying the architecture and lowering energy consumption. These design choices ensure that our framework is both energy-efficient and optimized for neuromorphic deployment.

## D. Implementation Details

The hyperparameters for SVL pretraining are presented in Tab. 17. The hyperparameters for SVL fine-tuning on 3D point clouds are detailed in Tab. 15, and those for SVL fine-tuning on DVS are outlined in Tab. 16.

## E. Limitations

This study also has several limitations, which simultaneously highlight promising avenues for further work. First, while SVL demonstrates substantial efficiency gains, our current evaluation is restricted to specific model scales. Validating the scalability of the proposed spike-driven mechanisms on larger-scale foundation models remains a crucial step to fully establish their applicability to more complex tasks.

*Table 14.* Hyperparameters for SVL pretraining.

| Architecture | E-3DSNN-T/S/L/H | Spike PointNet | Spike-driven PointFormer |
|---|---|---|---|
| Timestep (Training/Inference) | $1 \times 4/\ 4 \times 1$ | $1 \times 4/\ 4 \times 1$ | $1 \times 4/\ 4 \times 1$ |
| Epochs | 250 | 250 | 250 |
| Batch size | 4096 | 1024 | 1024 |
| Optimizer | AdamW | AdamW | AdamW |
| Base Learning rate | $2e-3$ | $2e-3$ | $3e-3$ |
| Learning rate decay | Cosine | Cosine | Cosine |
| Warmup eopchs | 10 | 10 | 10 |
| Weight decay | $1e-4$ | $1e-4$ | 0.1 |

*Table 15.* Hyper-parameters for SVL Finetuning on 3D point cloud.

| Hyper-parameter | ModelNet40 | ScanObjectNN | KITTI | SemanticKITTI |
|---|---|---|---|---|
| Timestep (Training/Inference) | $1 \times 4/\ 4 \times 1$ | $1 \times 4/\ 4 \times 1$ | $1 \times 4/\ 4 \times 1$ | $1 \times 4/\ 4 \times 1$ |
| Epochs | 300 | 250 | 100 | 80 |
| Batch size | 64 | 64 | 96 | 64 |
| Optimizer | AdamW | AdamW | AdamW | AdamW |
| Base Learning rate | $5e-4$ | $5e-3$ | $1e-2$ | $2e-3$ |
| Learning rate decay | Onecycle | Onecycle | Onecycle | Onecycle |

## F. Datasets

The ModelNet40 (Wu et al., 2015) dataset contains 12,311 CAD models across 40 object categories. Among them, 9,843 models are for training and 2,468 are for testing. The point clouds are clipped to ranges of $[-0.2m, 0.2m]$ for all X-, Y-, and Z-axes as the input data followed by voxelization with a resolution of $0.01m$. Classification performance was measured using overall accuracy metrics.

ScanObjectNN (Uy et al., 2019) consists of 11,416 training and 2,882 testing samples of real-world scanned 3D objects across 15 categories, with different degrees of data missing and noise contamination. The point clouds are clipped to ranges of $[-0.2m, 0.2m]$ for all X-, Y-, and Z-axes as the input data followed by voxelization with a resolution of $0.01m$.

The Objaverse dataset, which includes Objaverse-LVIS (Deitke et al., 2023) as a subset, is currently the largest 3D dataset. Objaverse-LVIS is a significant part of the Objaverse dataset, containing 46,832 annotated shapes across 1,156 LVIS categories. This extensive collection of 3D shapes provides a rich resource for researchers and practitioners in the field of computer vision and 3D modeling.

The large KITTI dataset (Geiger et al., 2012b) contains 7481 training samples, 3717 of which constitute trainsets and 3769 of which constitute validation sets. E-3DSNN is evaluated as backbones equipped with VoxelRCNN Head In detection (Deng et al., 2021). To execute our model, we uses OpenPCDet that is transformed into a spiking version by us. After being divided into regular voxels, raw point clouds are input to our 3DSNN on KITTI (Geiger et al., 2012a). The point clouds are clipped to ranges of $[-0.7m, 0.4m]$ for the X-axis, $[-40m, 40m]$ for the Y-axis, and $[-3m, 1m]$ for the Z-axis followed by voxelization with a resolution of $(0.05m, 0.05m, 0.1m)$. The Average Precision (AP) calculated by 11 recall positions for the Car class is used as the evaluation metrics.

The large SemanticKITTI dataset (Behley et al., 2019) contains 22 sequences from the raw KITTI dataset. About 1,000 lidar scans are included in each sequence, each of which corresponds to approximately 20,000 individual frames. We first adapted the Pointcept codebase into a spiking neural network (SNN) framework and utilized this customized implementation for model execution. Subsequently, we designed an asymmetric encoder-decoder architecture inspired by the UNet (Choy et al., 2019; Wu et al., 2023) paradigm, where the E-3DSNN acts as the encoder to extract hierarchical multi-scale features, while the decoder progressively fuses these features through skip connections to refine the output. During voxelize implementation, we set the window size to $[120m, 2°, 2°]$ for $(r, \theta, \phi)$. For data preprocessing, the input scene is restricted to the range $[-51.2m, -51.2m, -4m]$ to $[51.2m, 51.2m, 2.4m]$. The voxel size is set to $0.1m$.

The DVS Action dataset (Miao et al., 2019) comprises 10 actions performed by 15 subjects within 5s, which is recorded by DVS camera in an empty office. DVS is a vision sensor (Miao et al., 2019) that can records a sequence of tuples $[t, x, y, p]$

*Table 16.* Hyper-parameters for SVL Finetuning on temporal datasets.

| Hyper-parameter | Synthia 4D | DVS Action | DVS128 Gesture |
|---|---|---|---|
| Timestep (Training/Inference) | $1 \times 2/\ 3 \times 2$ | $1 \times 4/\ 6 \times 4$ | $1 \times 4/\ 6 \times 4$ |
| Epochs | 250 | 250 | 250 |
| Batch size | 64 | 64 | 64 |
| Optimizer | AdamW | AdamW | AdamW |
| Base Learning rate | $2e - 3$ | $2e - 3$ | $2e - 3$ |
| Learning rate decay | Cosine | Cosine | Cosine |
| Weight decay | $1e - 4$ | $1e - 4$ | $1e - 4$ |

*Table 17.* Hyper-parameters and training details for Spike-driven PointFormer with SVL on open-world multimodal learning.

| Hyper-parameter | Stage-1 (Feature Alignment) | Stage-2 (Instruction Tuning) |
|---|---|---|
| Optimizer | AdamW | AdamW |
| Learning rate decay | Cosine | Cosine |
| Epochs | 3 | 3 |
| Batch size | 128 | 32 |
| Base Learning rate | $2 \times 10^{-3}$ | $2 \times 10^{-5}$ |
| Weight decay | $1 \times 10^{-4}$ | $1 \times 10^{-4}$ |
| Dataset | 660K | 70K |

for each event streams. Among them, $t$ represents the timestamp of the event, $(x, y)$ represents the event's pixel coordinates and $p$ represents the polartity of the event.

The DVS128 Gesture dataset (Amir et al., 2017) contains 1,342 instances across 11 different hand and arm gestures, which are performed by 29 subjects under 3 distinct lighting conditions in 122 trials. They are captured by DVS128 camera, a DVS with $128 \times 128$ pixel resolution.

Synthia 4D. We employ the Synthia dataset (Ros et al., 2016) to construct 3D video sequences. Specifically, we use six driving scenarios across nine different weather conditions. Each scenario provides four stereo RGB-D images captured from the roof of a moving car. Depth images are back-projected into 3D space to generate 3D video sequences. For training, we use sequences 1–4, excluding the sunset, spring, and fog conditions; validation is conducted on sequence 5 under foggy weather; and testing is performed on sequence 6 under sunset and spring conditions. In total, the training, validation, and test splits contain 20,000, 815, and 1,886 3D scenes, respectively. Since the dataset is fully synthetic, we augment it with various types of noise to simulate realistic observations. These include elastic distortion, Gaussian noise, and chromatic shifts applied to the input point clouds.

MLLM training dataset. We further construct a large-scale point–text instruction-following dataset comprising approximately 730K samples and 60K instruct dataset following (Xu et al., 2024). This dataset is designed to support effective training by covering a broad spectrum of topics such as color, shape, usage, and material, thereby enabling robust multimodal instruction-following capabilities.

## G. Backpropagation process of I-LIF

There exist two primary methods of training high-performance SNNs. One way is to discretize ANN into spike form through neuron equivalence (Li et al., 2021b), i.e., ANN-to-SNN conversion, but this requires a long simulation time step and boosts the energy consumption. We employ the direct training method (Wu et al., 2018; Qiu et al., 2024) and apply surrogate gradient training.

Then in this section, we introduce the training process of SNN gradient descent and the parameter update method of spatio-temporal backpropagation (STBP) (Wu et al., 2018; Xiao et al., 2022; Hu et al., 2024). SNNs' parameters can be taught using gradient descent techniques, just like ANNs, after determining the derivative of the generation process.

Moreover, the accumulated gradients of loss $\mathcal{L}$ with respect to weights $\mathbf{w}$ at layer $\ell$ can be calculated as:

$$\frac{\partial \mathcal{L}}{\partial W^\ell} = \sum_{t=1}^{T} \frac{\partial \mathcal{L}}{\partial s^{\ell+1}[t]} \frac{\partial s^{\ell+1}[t]}{\partial u^{\ell+1}[t]} \left( \frac{\partial u^{\ell+1}[t]}{\partial w^\ell} + \sum_{\tau < t} \prod_{i=t-1}^{\tau} \left( \frac{\partial u^{\ell+1}[i+1]}{\partial u^{\ell+1}[i]} + \frac{\partial u^{\ell+1}[i+1]}{\partial s^{\ell+1}[i]} \frac{\partial s^{\ell+1}[i]}{\partial u^{\ell+1}[i]} \right) \frac{\partial u^{\ell+1}[\tau]}{\partial W^\ell} \right), \quad (20)$$

where $s^\ell[t]$ and $u^\ell[t]$ represent the binary and membrane potential of the neuron in layer $\ell$, at time $t$. Moreover, notice that $\frac{\partial s^\ell[t]}{\partial u^\ell[t]}$ is non-differentiable. To overcome this problem, (Wu et al., 2018) propose the surrogate function to make only the neurons whose membrane potentials close to the firing threshold receive nonzero gradients during backpropagation. In this paper, we use the rectangle function, which has been shown to be effective in gradient descent and may be calculated by:

$$\frac{\partial s^\ell[t]}{\partial u^\ell[t]} = \frac{1}{a} \operatorname{sign} \left( \left| u^\ell[t] - \vartheta \right| < \frac{a}{2} \right), \quad (21)$$

where $a$ is a defined coefficient for controlling the width of the gradient window, and is set to 1 in our paper.

## H. Architecture Details

In this section, we present the detailed architectural designs of E-3DSNN, Spike PointNet and Spike-driven PointFormer, outlining their core components, network configurations, and the specific adaptations made to enable efficient analysis of 3D point cloud within the spiking neural network framework.

**E-3DSNN (Qiu et al., 2025a)** are realized by adjusting the number of blocks and channels across stages to balance model size and performance. As shown in Tab. 18, the architecture scales from lightweight (E-3DSNN-T) to high-capacity (E-3DSNN-H) models, with corresponding changes in parameters and feature dimensions.

*Table 18.* Architecture details of E-3DSNN (Qiu et al., 2025a)

| Types | Blocks | Channels | Param. (M) |
|---|---|---|---|
| E-3DSNN-T | [1, 1, 1, 1] | [16, 32, 64, 128] | 1.8 |
| E-3DSNN-S | [1, 1, 1, 1] | [24, 48, 96, 160] | 3.2 |
| E-3DSNN-L | [2, 2, 2, 2] | [64, 128, 128, 256] | 17.3 |
| E-3DSNN-H | [2, 2, 2, 2] | [96,192,288,384] | 46.5 |

**Spike PointNet (Ren et al., 2024)** is the first spiking neural network specifically designed for efficient deep learning on 3D point clouds. It leverages the sparse and event-driven nature to achieve high accuracy with few parameters and low power consumption. This makes it particularly well-suited for deployment in energy-constrained or real-time 3D perception scenarios.

**Spike-driven PointFormer** is our proposed Transformer-based SNN backbone for point cloud encoding.

