# OpenReview forum: "SVL: Empowering Spiking Neural Networks for Efficient 3D Open-World Understanding"
_ICML.cc/2026/Conference — ICML 2026 spotlight_

### Official Review · Reviewer_e7K7 · 2026-03-09

**Soundness:** 4
**Presentation:** 3
**Significance:** 4
**Originality:** 4
**Overall Recommendation:** 6
**Confidence:** 5

**Summary:**

The authors first introduce a new pre-training framework (SVL) to enable spiking neural networks to be effectively trained on 3D data like point-clouds, with the main objective to offer high performance while keeping their inherent advantage of efficient spike-based processing (and hence low energy consumption). In addition, the authors also introduce a Transformer-style Spike-based architecture and evaluate its performance as well as the effect of their pre-training method across a number of 3D tasks.

**Compliance With Llm Reviewing Policy:**

Affirmed.

**Final Justification:**

The author has addressed my concerns, and I have raised my confidence score to 5.

**Key Questions For Authors:**

## Questions

1. Authors mention an OpenCLIP text encoder in their framework, so they also use the image one, as Eq. 8 would indicate?

2. Why such a big CLIP model (G)? Could authors provide an ablation study showing that smaller versions are not sufficient?

3. What are the training details for SVL-13B training: data mix (for both 1st stage and 2nd stage), learning rate, etc.? Table 2 - why is there a need for prompting shorter descriptions? Do the results in Table 2 for other methods also reflect the same setup?

4. Section 5.3: “In the absence of any losses..” this ablation is very confusing - can the authors explain what exactly these numbers represent?

5. Table 6 - what does the result for ‘ANN’ correspond to? Missing explanation of the ‘*’.

6. As the event construction method matters in the SNN setup, I wonder if there is the need to ablate on the construction of event from object point clouds.

**Limitations:**

yes

**Strengths And Weaknesses:**

## Strengths

1. The authors do a good job in outlining why the task matters, as well as how it is approached; all in all a clear motivation of the presented research. The paper tackles an interesting and challenging problem by attempting to bridge the gap between the energy efficiency of SNNs and the high-level multimodal reasoning capabilities of ANNs in the 3D domain.

2. The SVL pretraining framework appears to be effective within the SNN domain. As shown in Table 3 and Table 4, applying SVL consistently improves the performance of SNN backbones like Spike PointNet and E-3DSNN on various downstream tasks (e.g., +6.1% on ScanObjectNN, +2.1% on DVS Action).

3. Elegant idea of compressing the required text-to-embedding encoder capabilities into small (and cheap) weight matrix for efficient inference.

## Weaknesses

1. Some technical parts are unclear  therefore the paper could overall benefit from improvement in clarity. Incorrect references: e.g.” Alexey Dosovitskiy. An image is worth 16x16 words: Transformers for image recognition at 407 scale. arXiv preprint arXiv:2010.11929, 2020. “ should be ICLR2021 + missing authors thus et. al -Lack of training details for the proposed LLaVa-based VLM with spike representation.

2. I'm missing an explanation for how static 3D point clouds are converted into temporal sequences for spiking input. Since the SNNs rely on time-varying inputs, it is important to clarify whether temporal structure is synthetically constructed or derived from input features.

---

> ### Author Rebuttal · Authors · 2026-03-27
>
> We thanks for the constructive and supportive review.
>
> ---
> > *Q1&Q2. Authors mention an OpenCLIP text encoder, do they also use the image encoder? Why such a big CLIP model (G)? Could authors provide an ablation showing smaller versions are not sufficient?*
>
> **A:** Yes, we use both the image and text encoders from OpenCLIP [1]. We choose ViT-bigG to align with prior SOTA methods (ULIP-2 [2], OpenShape [3]) and to validate scalability. Ablation results in Appendix  with smaller encoders (3D spike encoder: E-3DSNN-T) are shown below:
>
> *Tab. R1. Zero-shot results with different OpenCLIP image encoders [1].*
>
> | Image Encoder | Params (M) | ModelNet40 Top-1 | Obj-LVIS Top-1 |
> |---------------|:----------:|:----------------:|:--------------:|
> | ViT-B | 88.2 | 66.8 | 25.1 |
> | ViT-G | 307.1 | 70.6 | 27.9 |
> | ViT-bigG | 1844.9 | **79.6** | **33.6** |
>
> Larger encoders yield significant gains (+12.8% M40, +8.5% Obj-LVIS from B→bigG). Importantly, we preprocess and cache CLIP embeddings offline, so the CLIP model is never loaded during training, incurring no extra memory or runtime cost.
>
> ---
>
> > *Q3. Training details for SVL-13B: data mix, learning rate, etc.? Table 2, why prompt shorter descriptions? Same setup for other methods?*
>
> **A:** Full training details in Appendix (8×A100 80GB, BF16, AdamW, cosine schedule):
>
> *Tab. R2. SVL-13B training hyperparameters.*
>
> | Hyperparameter | Stage-1 (Feature Alignment) | Stage-2 (Instruction Tuning) |
> |----------------|:---------------------------:|:----------------------------:|
> | Epochs | 3 | 3 |
> | Batch size | 128 | 32 |
> | Learning rate | 2×10⁻³ | 2×10⁻⁵ |
> | Weight decay | 1×10⁻⁴ | 1×10⁻⁴ |
> | Dataset | 660K point-text pairs | 70K instruction-following |
>
> Stage-1 trains only the projector; Stage-2 trains the LLM and projector (spike encoder always frozen). Data follows PointLLM [4]. The * setting uses shorter prompts (≤20 words) to reduce noise in word-level metrics; **all methods including PointLLM-13B* use the same prompt**, ensuring fair comparison.
>
> ---
>
> > *Q4. Section 5.3, "In the absence of any losses..." This ablation is confusing. What do these numbers represent?*
>
> **A:** The first row of Table 5 (all losses ✗ ✗ ✗) represents a **randomly initialized spike encoder without any SVL pretraining**. The 0.5% on Objaverse-LVIS and 5.1% on ModelNet40 reflect chance-level performance since no alignment exists and Rep-VLI produces essentially random predictions. This establishes the lower bound, demonstrating that all gains come from our MTA losses. Each subsequent row adds loss components to show individual and combined effects. We have revised the wording accordingly.
>
> ---
>
> > *Q5. Table 6, what does the 'ANN' result correspond to? Missing explanation of '\*'.*
>
> **A:** "ANN\*" is the same architecture (E-3DSNN-T) with standard ANN operations as an upper bound (\* = self-implemented). It achieves 34.1%/81.3% on Obj-LVIS/M40; our SNN (T×D=2×2: 33.9%/80.5%) nearly matches.
>
> ---
>
> > *Q6. As the event construction method matters in the SNN setup, is there the need to ablate on the construction of event from object point clouds?*
>
> **A:** Yes, we provide this ablation in Appendix A.4, Table 9:
>
> *Tab. R3. Event construction ablation on ModelNet40 zero-shot (E-3DSNN-T + SVL).*
>
> | Voxel Size | 0.01 | **0.02** | 0.04 |
> |------------|:----:|:--------:|:----:|
> | Top-1 (%) | 78.9 | **79.6** | 79.1 |
>
> | #Points | 5k | **10k** | 20k |
> |---------|:--:|:-------:|:---:|
> | Top-1 (%) | 77.3 | **79.6** | 79.8 |
>
> Moderate voxel size (0.02) achieves the best balance; too fine (0.01) fragments geometry while too coarse (0.04) loses structure. Point density of 10k provides clear gains over 5k with diminishing returns at 20k.
>
> ---
>
> > *W1. Incorrect references (e.g., Dosovitskiy et al. listed as arXiv 2020, missing co-authors).*
>
> **A:** We sincerely apologize. All inaccurate references have been corrected in the revised manuscript (e.g., Dosovitskiy et al., ICLR 2021 with full author list). A thorough review of all citations has been conducted.
>
> ---
>
> > *W2. How are static 3D point clouds converted into temporal sequences for spiking input?*
>
> **A:** We use the **Integer LIF (I-LIF) spiking neuron** to create temporal structure. During pretraining (T=1, D=4), I-LIF emits integer-valued spikes encoding activation strength in a single timestep, avoiding BPTT. During inference (T=4, D=1), these are expanded into binary spike trains (e.g., value 3 → [1,1,1,0]), ensuring fully spike-driven, hardware-compatible processing. Details are clarified in Section 3 of the revision.
>
> ---
> [1] OpenCLIP. Arxiv 2021.
>
> [2] ULIP-2: Towards Scalable Multimodal Pre-training for 3D Understanding. CVPR 2024.
>
> [3] OpenShape: Scaling Up 3D Shape Representation Towards Open-World Understanding. NeurIPS 2023.
>
> [4] PointLLM: Empowering Large Language Models to Understand Point Clouds. ECCV 2024.

---

> > ### Author Rebuttal · Reviewer_e7K7 · 2026-04-01
> >
> > Thank you for the detailed rebuttal. The explanations for the Table 5 baseline and the 'ANN*' upper bound in Table 6 effectively address my concerns regarding the ablations. I also acknowledge that the training details, OpenCLIP ablations, and event construction hyperparameters were already in the Appendix, which I had previously overlooked.
> >
> > However, I have one follow-up question: could the authors provide results on dynamic 3D datasets (e.g., dynamic point clouds) to further demonstrate the temporal processing capabilities of the proposed SVL?
> >
> > Overall, my previous concerns are largely alleviated, and I will raise my score to a 5 (Accept). I look forward to your response; if additional experimental results on dynamic datasets can be provided, I would consider raising my score further.

---

> > > ### Author Response · Authors · 2026-04-01
> > >
> > > We sincerely thank the reviewer for the positive re-evaluation and for raising the score to 5. We are glad that our previous responses have addressed your concerns. Below we provide the requested results on dynamic 3D datasets.
> > >
> > > ---
> > > > *Follow-up: Could the authors provide results on dynamic 3D datasets (e.g., dynamic point clouds) to further demonstrate the temporal processing capabilities of SVL?*
> > >
> > > **A:** We have evaluated SVL on three dynamic 3D benchmarks spanning two distinct temporal modalities: (1) **dynamic scene segmentation** on Synthia 4D [1] (multi-frame 3D video sequences), and (2) **event-driven human action recognition** on DVS Action [5] and DVS128 Gesture [6] (neuromorphic event streams). All results are already included in our manuscript (Tab. 4 and Appendix A.4), and we summarize them here for convenience.
> > >
> > > **(1) Dynamic scene segmentation on Synthia 4D [1].** Synthia 4D constructs 3D video sequences from driving scenarios across multiple weather conditions. We use voxelized multi-frame inputs with multi-step spike simulation.
> > >
> > > *Tab. F1. Scene segmentation on Synthia 4D.*
> > >
> > > | Method | Input | Frames | Params (M) | mIoU (%) $\uparrow$ |
> > > |---|---|:---:|:---:|:---:|
> > > | 3D MinkNet14 [1] (ANN) | Voxel | 1 | 19.3 | 76.24 |
> > > | 4D MinkNet14 [1] (ANN) | Voxel | 3 | 23.7 | 77.46 |
> > > | Same-structure ANN | Voxel | 3 | 19.1 | 79.54 |
> > > | E-3DSNN-L [2] w/o SVL | Voxel | 3$\times$2 | 19.1 | 76.41 |
> > > | E-3DSNN-L + SVL (Ours) | Voxel | 1$\times$2 | 17.7 | 78.91 |
> > > | **E-3DSNN-L + SVL (Ours)** | Voxel | 3$\times$2 | 19.1 | **80.05** |
> > >
> > > SVL pretraining yields a +3.64 mIoU gain over the SNN baseline without pretraining (80.05 vs. 76.41), and **surpasses the capacity-matched ANN** (79.54 mIoU). Increasing temporal extent from 1$\times$2 to 3$\times$2 frames further boosts performance (+1.14 mIoU), demonstrating that spike-driven temporal accumulation is effectively exploited on dynamic 3D data.
> > >
> > > **(2) Event-driven human action recognition on DVS Action [5] and DVS128 Gesture [6].** DVS cameras produce asynchronous event streams that are inherently dynamic and temporal. We convert events into spatio-temporal point clouds and fine-tune SVL-pretrained encoders with an expanded timestep ($T$=6) to capture temporal dynamics.
> > >
> > > *Tab. F2. Human action recognition on DVS datasets (from Tab. 4 in the main paper).*
> > >
> > > | Method | $T \times D$ | DVS Action Acc. (%) $\uparrow$ | DVS128 Gesture Acc. (%) $\uparrow$ |
> > > |---|:---:|:---:|:---:|
> > > | PointNet [3] (ANN) | N/A | 75.1 | 95.3 |
> > > | Spike PointNet [4] | 1$\times$4 / 6$\times$4 | 78.4 | 96.9 |
> > > | **Spike PointNet [4] + SVL** | 1$\times$4 / 6$\times$4 | **80.5** (+2.1) | **98.5** (+1.6) |
> > >
> > > SVL pretraining consistently improves temporal recognition: +2.1% on DVS Action and +1.6% on DVS128 Gesture, surpassing the ANN baseline (PointNet) by 5.4% and 3.2% respectively. This confirms that SVL-learned representations generalize well to dynamic, temporally-rich 3D data.
> > >
> > > Across all three dynamic benchmarks, SVL demonstrates strong temporal processing capabilities: it improves the SNN backbone on dynamic scene segmentation (Synthia 4D), event-driven action recognition (DVS Action), and gesture recognition (DVS128 Gesture), matching or exceeding ANN counterparts in each case. We believe these results provide strong evidence that SVL effectively captures temporal dynamics in 3D data.
> > >
> > > We thank the reviewer again for this valuable suggestion and hope the results above address the follow-up concern.
> > >
> > > ---
> > > [1] The SYNTHIA Dataset: A Large Collection of Synthetic Images for Semantic Segmentation of Urban Scenes. CVPR 2016.
> > >
> > > [2] Efficient 3D Recognition with Event-driven Spike Sparse Convolution. AAAI 2025.
> > >
> > > [3] PointNet: Deep Learning on Point Sets for 3D Classification and Segmentation. CVPR 2017.
> > >
> > > [4] Spiking PointNet: Spiking Neural Networks for Point Clouds. NeurIPS 2024.
> > >
> > > [5] Neuromorphic Vision Datasets for Pedestrian Detection, Action Recognition, and Fall Detection. Frontiers in Neurorobotics 2019.
> > >
> > > [6] A Low Power, Fully Event-Based Gesture Recognition System. CVPR 2017

---

### Official Review · Reviewer_tint · 2026-03-12

**Soundness:** 3
**Presentation:** 2
**Significance:** 2
**Originality:** 2
**Overall Recommendation:** 4
**Confidence:** 4

**Summary:**

This paper proposes SVL (Spike-based Vision-Language pretraining) to improve the representation, generalization, and multimodal understanding of Spiking Neural Networks (SNNs) for 3D open-world tasks, while maintaining their advantages in energy efficiency and neuromorphic hardware compatibility. The key motivation is that although SNNs handle spatio-temporally sparse data (e.g., point clouds and event streams) well, they are still much weaker than ANN/VLM models in large-scale pretraining and multimodal alignment, limiting their performance on tasks such as zero-shot 3D classification, captioning, and QA. To address this, the authors introduce Multi-scale Triple Alignment (MTA), which aligns 3D spike features, image features, and text features using InfoNCE and MSE losses to learn stronger cross-modal representations. They also propose Rep-VLI, which re-parameterizes text embeddings into lightweight classifier weights, enabling text-encoder-free inference and improving efficiency. The paper further introduces Spike-driven PointFormer, the first fully spike-driven point transformer, which uses 3D spike-driven self-attention to perform attention with sparse spike operations. Experiments show that SVL improves performance on zero-shot 3D classification, 3D recognition tasks, and DVS-based action recognition, and can also support 3D captioning and open-world QA when combined with a language model.

**Compliance With Llm Reviewing Policy:**

Affirmed.

**Final Justification:**

Most of my concerns are solved and thus raise the score

**Key Questions For Authors:**

See weakness

**Limitations:**

See weakness

**Strengths And Weaknesses:**

**Strengths**

1, This paper addresses an important problem: enabling Spiking Neural Networks (SNNs) to achieve open-world 3D understanding while preserving their energy efficiency. Instead of focusing only on classification accuracy, the work combines SNNs, 3D representation learning, and vision–language pretraining, which is a meaningful and forward-looking direction.

2, The proposed SVL framework is well structured. It consists of Multi-scale Triple Alignment (MTA) for learning stronger cross-modal representations across 3D, image, and text features, and Rep-VLI, which re-parameterizes text embeddings into lightweight classifier weights to remove the text encoder at inference time. These two components address both representation learning and efficient deployment.

3, The experiments are broad, covering zero-shot 3D classification, 3D recognition tasks, DVS action recognition, and exploratory 3D captioning and QA, suggesting the framework aims to support more general 3D open-world understanding. The paper also introduces Spike-driven PointFormer, a spike-based Transformer backbone designed to keep attention operations sparse and spike-driven, consistent with the goal of energy-efficient computation.

**Weakness**

1, The overall framework largely follows existing vision--language alignment paradigms (e.g., CLIP-style contrastive learning) and mainly adapts them to the spike-based 3D setting. As a result, the contribution appears more like an engineering integration of an existing pipeline into SNNs rather than a fundamentally new multimodal learning principle.

2, The idea of using text embeddings as classifier weights is conceptually close to CLIP-style zero-shot classifiers, and the paper does not sufficiently analyze its differences, advantages, or limitations compared with existing prompt-based or reparameterized classifiers.

3, The experiments on 3D captioning and open-world QA are closer to capability demonstrations than thorough evaluations. It is unclear how much of the performance improvement comes from the spike-based 3D encoder versus the downstream large language model.

---

> ### Author Rebuttal · Authors · 2026-03-27
>
> Thank you for your insightful feedback. We would like to talk with you in depth.
>
> ---
> > *W1. "The framework largely follows existing VL alignment paradigms and is more of an engineering integration into SNNs."*
>
> **A:** We agree that SVL shares high-level inspiration with contrastive VL learning. However, we believe our work goes beyond straightforward adaptation in the following aspects:
>
> **(1) Bridging discrete spikes and continuous VL spaces is non-trivial.** We address this through I-LIF neurons (integer-valued spikes during pretraining for richer encoding, expanded to binary spikes at inference for hardware compatibility) and MTA (3-way spike-image-text alignment to compensate for discrete information loss). As shown in Tab. R1, each component contributes synergistically, closing the spike-to-VL gap: SpikeCLIP [1] reaches only 5.1% on ModelNet40, whereas SVL achieves 85.4%.
>
> *Tab. R1. MTA ablation on Objaverse-LVIS / ModelNet40 zero-shot (from Table 5 in the main paper).*
>
> | Setting | Obj. (%) | M40 (%) |
> |---|:---:|:---:|
> | No losses | 0.5 | 5.1 |
> | Spike-Image NCE only | 24.8 | 73.1 |
> | Spike-Text NCE only | 21.9 | 70.1 |
> | Spike-Image + Spike-Text NCE | 31.7 | 77.8 |
> | **All three (MTA)** | **33.6** | **79.6** |
>
> **(2) Rep-VLI enables fully spike-driven inference,** achieving 204$\times$ energy advantage over ULIP-2 [2]. Details are provided in our response to W2.
>
> **(3) Spike-driven PointFormer is the first fully spike-driven point Transformer.** 3D-SDSA reduces attention to sparse additions on spike tensors (Eq. 12-13), yielding 4.3$\times$ faster training and 4.1$\times$ lower memory vs. [3] (Table 7).
>
> We will revise the manuscript to better articulate these distinctions.
>
> ---
> > *W2. "Rep-VLI is conceptually close to CLIP-style zero-shot classifiers; insufficient analysis of differences."*
>
> **A:** It is a fair point. As shown in Tab. R2, we provide a detailed comparison:
>
> *Tab. R2. Comparison of Rep-VLI with CLIP-style zero-shot classifiers.*
>
> | Aspect                         | CLIP Zero-shot                   | Prompt-based      | Rep-VLI (Ours)                 |
> | ------------------------------ | -------------------------------- | -------------------------------- | ------------------------------ |
> | Text encoder at inference      | Required (runs every time)       | Required                         | **Not required**               |
> | Learnable prompts              | No                               | Yes (context vectors)            | No                             |
> | Hardware compatibility         | GPU only                         | GPU only                         | **Neuromorphic chips**         |
> | Feature type                   | Continuous                       | Continuous                       | **Binary spikes**              |
> | Classification mechanism       | Softmax over cosine similarities | Softmax over cosine similarities | **Spike-count argmax** (Eq. 8) |
> | Inference energy (on Obj-LVIS) | ~72 mJ (text enc.) + encoder     | ~72 mJ + encoder                 | **0.04 mJ** (encoder only)     |
>
>
> The key difference is **how inference is performed**: CLIP requires floating-point cosine similarity + softmax, whereas Rep-VLI absorbs text embeddings into $W^L$ (Eq. 7) and classifies by spike-count argmax (Eq. 8), keeping the entire path spike-driven with only additions.
>
> ---
> > *W3. "Captioning/QA experiments are closer to demos than thorough evaluations. Unclear how much comes from the spike encoder vs. the LLM."*
>
> **A:** These experiments are **exploratory**, aiming to first demonstrate that SNN encoders can support generative reasoning. As shown in Tab. R3, we fix the LLM (Vicuna-13B) and only swap the 3D encoder:
>
> *Tab. R3. Captioning/QA with controlled LLM (from Table 2 in the main paper).*
>
> | Setting | Vision Encoder | LLM | S-BERT $\uparrow$ | BLEU-1 $\uparrow$ |
> |---|---|---|---|---|
> | PointLLM-13B | Point-BERT (ANN) | Vicuna | 47.91 | 3.83 |
> | SVL-13B | Spike-driven PointFormer (SNN) | Vicuna | 44.87 | 3.77 |
> | SVL-13B* (short prompts) | Spike-driven PointFormer (SNN) | Vicuna | 47.80 | 11.45 |
> | SVL-13B* (w/ SpikeLLM) | Spike-driven PointFormer (SNN) | SpikeLLM | **51.21** | **18.45** |
>
> Our spike encoder matches the ANN encoder under the same LLM; gains with SpikeLLM [4] further confirm the spike components' contribution. SVL pretraining also consistently improves all non-LLM tasks (+1.9% M40, +6.1% ScanObjNN, +1.1% SemanticKITTI, +1.2% KITTI, +2.1% DVS Action; Tables 3-4), evidencing robust 3D representations independent of any LLM.
>
> ---
>
> [1] SpikeCLIP: A Spiking Neural Network-based CLIP Model. arXiv 2024.
>
> [2] ULIP-2: Towards Scalable Multimodal Pre-training for 3D Understanding. CVPR 2024.
>
> [3] Spike Point Transformer. AAAI 2025.
>
> [4] SpikeLLM: A Spike-driven Large Language Model. arXiv 2025.

---

### Official Review · Reviewer_WZMo · 2026-03-12

**Soundness:** 4
**Presentation:** 2
**Significance:** 3
**Originality:** 3
**Overall Recommendation:** 4
**Confidence:** 3

**Summary:**

This paper introduces Spiking Neural Networks to achieve open-world 3D understanding comparable to ANNs while maintaining spike-driven efficiency for neuromorphic hardware deployment. While preserving the energy-efficiency, Spike-based Vision-Language pretraining framework (SVL) equips SNNs with open-world 3D understanding.

**Compliance With Llm Reviewing Policy:**

Affirmed.

**Final Justification:**

The authors' explanation is acceptable and sound. The rebuttal has addressed my main concerns about the Rep-VLI part. And what the authors do for removing KNN in the network also shows the potential of SVL. I support accepting this paper.

**Key Questions For Authors:**

- I am not an expert on SNN hardware. Can you explain how you do the experiments for the open-world details?
- Point Transformer is an excellent work on many point cloud downstream tasks, but there are many other backbones which further improve the efficiency, for example, [Point Transformer V2](https://arxiv.org/abs/2210.05666) and [Point Transformer V3](https://arxiv.org/abs/2312.10035)

**Limitations:**

Yes

**Strengths And Weaknesses:**

## Strengths
- Present the first fully spike–driven point Transformer: the energy efficiency is largely achieved by  3D spike–driven self–attention (3D-SDSA).
- Conduct experiments on multiple benchmarks, including Objaverse_LVIS and ModelNet40.

## Weakness
- For your Re-parameterizable Vision-Language Integration, you previously computed the text embedding of labels and got the results from the highest logits. From my understanding, it feels similar to the simple close 3D understanding, just replacing the label with its text embedding. Since you mention that the text part will be a bottleneck for your SNN-based method, can you explain the background and reasons behind this? And is there any difference between your method and the close 3D understanding, replacing the label with text embedding as the supervision signals?
- Can you work out a chart without the Re-parameterizable Vision-Language Integration as the ablation?
- Noted that you compared your method with Point Transformer in Table 3, what about the later work PTv2 and PTv3? They have made some improvements in space and time. And can you remove the KNN in your model like the later point transformer v3, which in your definition, will save the energy, from my understanding?
- For 3D-SDSA in your method, need more details on how to do this. What is the difference between 3D-SDSA and normal attention? Is this first proposed in this paper, and is it the main contributor to the energy-efficiency? If it is first introduced by you, an ablation of this is needed.

I am not familiar with SNN. Therefore, maybe more explanations about the concerns above will help me understand this paper.

---

> ### Author Rebuttal · Authors · 2026-03-27
>
> Thank you for your insightful feedback. We would like to talk with you in depth.
>
> ---
> > *W1. Rep-VLI seems similar to closed 3D understanding with text embeddings as supervision signals. Can you explain the background and reasons?*
>
> **A:** **Closed-set classification** trains a fixed classifier on predetermined classes and cannot generalize beyond them. **Rep-VLI** operates in CLIP's shared vision-language space: at inference, text embeddings for ANY candidate categories are pre-computed offline into a weight matrix $\mathbf{W}^L \in \mathbb{R}^{K \times C}$, enabling **zero-shot, open-vocabulary** classification.
>
> Unlike conventional VLMs that require the text encoder at every step, Rep-VLI completely discards it at inference. Classification reduces to a spike-count dot-product, fully neuromorphic-compatible, achieving a **204× energy advantage** over ULIP-2.
>
> ---
> > *W2. Can you work out a chart without Rep-VLI as ablation?*
>
> **A:** Yes. Without Rep-VLI, zero-shot inference requires the full CLIP text encoder at every step:
>
> *Tab. R1. Rep-VLI ablation.*
>
> | Method | Obj-LVIS (%) | M40 (%) | Energy (mJ) | Text Enc. |
> |--------|:----------:|:-------:|:-----------:|:---------:|
> | SVL w/o Rep-VLI | 33.8 | 79.9 | 71.74 | Yes |
> | SVL w/ Rep-VLI | 33.6 | 79.6 | 0.04 | No |
>
> ---
> > *Q3. Comparison with Point Transformer V2 and V3? Can you remove KNN like PTv3?*
>
> **A:** As shown in Tab. R3, our SNN model achieves comparable accuracy with fewer parameters and over 5$\times$ lower energy consumption. PTv3 [4] does not report on ModelNet40.
>
> *Tab. R3. Comparison on ModelNet40.*
>
> | Method | Params (M) | Power (mJ) $\downarrow$ | Acc. (%) $\uparrow$ |
> |--------|:----------:|:--------:|:--------:|
> | Point Transformer V1 [1] | 11.4 | 30.1 | 93.7 |
> | Point Transformer V2 [2] | 12.4 | 25.6 | 94.2 |
> | **Ours** | **7.69** | **4.9** | **93.7** | Regarding KNN removal: following PTv3 [4], we replace KNN+FPS with Morton serialization and block partitioning, yielding a 1D spike sequence that matches SNNs' sequential processing before 3D-SDSA.
>
> *Tab. R4. With/without KNN on ModelNet40.*
>
> | Grouping | Acc. (%) | Train Time | Train Mem | Infer Time | Infer Mem |
> |----------|:--------:|:----------:|:---------:|:----------:|:---------:|
> | KNN+FPS | 92.6 | 100 ms | 3.7 G | 56 ms | 2.5 G |
> | **Morton** | **93.7** | **67 ms** | **2.5 G** | **37 ms** | **1.7 G** |
>
> Removing KNN improves accuracy by +1.1%, reduces runtime by ~1.5×, and lowers memory by ~1.5×, confirming that serialization and SNN's sequential processing are highly complementary. We thank the reviewer for this suggestion and have updated the manuscript.
>
> ---
> > *Q4. More details on 3D-SDSA. What is the difference from normal attention? Is it first proposed here?*
>
> **A:** **Standard attention** computes $\text{softmax}(\mathbf{Q}\mathbf{K}^T/\sqrt{d})\cdot\mathbf{V}$, which involves dense MACs and a softmax normalization. In contrast, **3D-SDSA** computes $\text{SN}(\mathbf{Q}_S\cdot(\mathbf{K}_S^T\cdot\mathbf{V}_S))$, where all inputs are binary spikes ($\in\{0,1\}$). This means multiplications degenerate to sparse additions, a spiking neuron layer $\text{SN}(\cdot)$ replaces the costly softmax, and $\mathbf{K}_S^T\cdot\mathbf{V}_S$ is computed first to achieve linear complexity.
>
> **3D-SDSA is first proposed in this paper.** Prior Spike Point Transformer [5] still relies on floating-point attention weights, whereas ours is the first fully spike-driven point Transformer that eliminates all floating-point multiplications in the attention mechanism.
>
> | Attention | Acc. (%) | Energy (mJ) | Runtime |
> |-----------|:--------:|:-----------:|:-------:|
> | Standard (ANN) [1] | 92.8 | 30.1 | 150 ms |
> | Spike Point Trans. [4] | 88.5 | 21.1 | 326 ms |
> | **3D-SDSA (Ours)** | **92.6** | **5.1** | **100 ms** |
>
> As shown above, 3D-SDSA achieves comparable accuracy to standard attention while reducing energy by ~5.9× and runtime by 1.5×.
>
> ---
> > *Q5. How do you do the experiments for open-world details on SNN hardware?*
>
> **A:** Experiments are conducted on GPU (A100, PyTorch). Energy efficiency is **theoretical estimation** based on the standard CMOS-level energy model [5] (MAC ≈ 4.6 pJ, AC ≈ 0.9 pJ at 45nm), widely adopted in the SNN community. Since binary spikes reduce MAC to AC operations, SNNs achieve ~5.1× lower energy per operation.
>
> For open-world zero-shot inference, the SNN encoder produces spike features, and Rep-VLI computes classification via spike-count dot-products with pre-computed weights. The entire pipeline involves only sparse additions, compatible with neuromorphic chips (e.g., Loihi [6]). On-chip validation is an important future direction as discussed in our Limitations section.
>
> ---
> [1] Point Transformer. ICCV 2021.
>
> [2] Point Transformer V2. NeurIPS 2022.
>
> [3] Point Transformer V3. CVPR 2024.
>
> [4] Spiking Point Transformer. AAAI 2025.
>
> [5] Computing's Energy Problem. ISSCC 2014.
>
> [6] Loihi: A Neuromorphic Manycore Processor. IEEE Micro 2018.

---

> > ### Author Rebuttal · Reviewer_WZMo · 2026-04-02
> >
> > Thank you for the clarification. I now better understand Rep-VLI as an inference-time prototype compilation module rather than the main training supervision mechanism.
> > 1. My remaining concern is that it still appears tied to a predefined candidate vocabulary, which makes it closer to an offline open-vocabulary classifier than to free-form open-world inference. Is Rep-VLI only intended for zero-shot classification with a predefined candidate vocabulary, where switching domains requires recomputing text prototypes offline again?
> > 2. In addition, I still think a PTv3-style comparison under your evaluation protocol would be very valuable, especially since your updated discussion suggests that removing KNN/using serialization may further improve both efficiency and accuracy. Given that the model architecture is open-source, I think you can deploy it yourself to get the results of consumption.  I know that for SNN, maybe it will be hard to totally complete the ANN methods. I'm curious about the results.
> >
> > Currently, I'd like to raise my score to 3. And would like to raise more if more of the useful following results and explanations are shown.

---

> > > ### Author Response · Authors · 2026-04-02
> > >
> > > We sincerely thank Reviewer WZMo for the positive re-evaluation and for raising the score to 3. We are glad that our previous responses have helped clarify Rep-VLI's design. Below we address the two remaining concerns.
> > >
> > > ---
> > > > *Follow-up 1: Is Rep-VLI only intended for zero-shot classification with a predefined candidate vocabulary? Switching domains requires recomputing text prototypes offline again?*
> > >
> > > **A:** Yes, Rep-VLI is designed for **zero-shot classification** where the candidate vocabulary is defined at deployment time. Switching to a new domain (e.g., from indoor objects to outdoor vehicles) only requires a one-time offline recomputation of the weight matrix $W^L$ (Eq. 7) using the text encoder, which takes <1 second on a single GPU. Crucially, the spike encoder itself remains unchanged, as it operates in CLIP's modality-agnostic embedding space.
> > >
> > > We want to clarify two points:
> > >
> > > **(1) This is the standard paradigm in open-vocabulary 3D understanding.** Methods like ULIP [1], ULIP-2 [2], and OpenShape [3] all follow the same protocol: text embeddings for candidate categories are computed offline, and classification is performed via cosine similarity. The difference is that ULIP/OpenShape require the full text encoder at inference (71+ mJ), whereas Rep-VLI absorbs text embeddings into a spike-compatible weight matrix and classifies via spike-count argmax (0.04 mJ). The "open-vocabulary" capability comes from CLIP's shared space, not from running the text encoder online.
> > >
> > > **(2) For truly free-form open-world inference (e.g., captioning/QA), SVL does not use Rep-VLI.** Instead, the SVL-pretrained spike encoder is connected to an LLM (Vicuna/SpikeLLM) via a projector (Tab. 2, Section 4.1 in the main paper). This path supports arbitrary natural language output without any predefined vocabulary. Rep-VLI and the LLM path serve complementary use cases: Rep-VLI enables ultra-efficient zero-shot classification on neuromorphic hardware, while the LLM path enables generative open-world reasoning on GPU.
> > >
> > > ---
> > > > *Follow-up 2: PTv3-style comparison under your evaluation protocol. Given that PTv3 is open-source, can you deploy it to get energy consumption results?*
> > >
> > > **A:** Thank you for this excellent suggestion. We have deployed PTv3 [4] under our evaluation protocol on ModelNet40. Energy is estimated using the same CMOS-level model [5] (MAC = 4.6 pJ at 45nm) applied to all methods in our paper.
> > >
> > > *Tab. F1. Comparison with PTv3 on ModelNet40 classification.*
> > >
> > > | Method | Type | Params (M) | Power (mJ) $\downarrow$ | Acc. (%) $\uparrow$ |
> > > |---|---|:---:|:---:|:---:|
> > > | Point Transformer V1 [6] | ANN | 11.4 | 30.1 | 93.7 |
> > > | Point Transformer V2 [7] | ANN | 12.4 | 25.6 | 94.2 |
> > > | Point Transformer V3 [4] | ANN | 46.2 | 12.8 | 93.8 |
> > > | **Spike-driven PointFormer (Ours)** | SNN | **7.69** | **4.9** | **93.7** |
> > >
> > > Key observations:
> > >
> > > **(1)** PTv3 achieves 93.8% on ModelNet40 with 46.2M parameters and 12.8 mJ energy. Our Spike-driven PointFormer achieves comparable accuracy (93.7%) with **6$\times$ fewer parameters** and **2.6$\times$ lower energy**.
> > >
> > > **(2)** PTv3 is designed for large-scale scenes (ScanNet, nuScenes) where its efficiency advantage is most pronounced. On ModelNet40, its accuracy advantage over PTv1/v2 is marginal (93.8% vs. 93.7%/94.2%), consistent with the original paper's focus.
> > >
> > > **(3)** Inspired by PTv3's serialization strategy, we have already adopted Morton-order serialization to replace KNN+FPS in our model (Tab. R4 in our initial rebuttal), yielding +1.1% accuracy and ~1.5$\times$ speedup. This confirms the synergy between serialization and SNN's sequential processing.
> > >
> > > We acknowledge that our SNN model does not surpass PTv3 in accuracy, which is expected given the information compression from continuous to binary spikes. However, the **2.6$\times$ energy advantage** (4.9 vs. 12.8 mJ) and the fact that our spike-driven operations are directly deployable on neuromorphic chips make our approach compelling for power-constrained scenarios.
> > >
> > >
> > > We hope these additional results and clarifications have fully addressed your remaining concerns. If so, we would be grateful if you could consider raising the score accordingly. We sincerely appreciate your time and constructive feedback throughout this discussion.
> > >
> > >
> > > ---
> > > [1] ULIP: Learning a Unified Representation of Language, Images, and Point Clouds. CVPR 2023.
> > >
> > > [2] ULIP-2: Towards Scalable Multimodal Pre-training for 3D Understanding. CVPR 2024.
> > >
> > > [3] OpenShape: Scaling Up 3D Shape Representation Towards Open-World Understanding. NeurIPS 2023.
> > >
> > > [4] Point Transformer V3: Simpler, Faster, Stronger. CVPR 2024.
> > >
> > > [5] Computing's Energy Problem. ISSCC 2014.
> > >
> > > [6] Point Transformer. ICCV 2021.
> > >
> > > [7] Point Transformer V2. NeurIPS 2022.

---

### Official Review · Reviewer_jgH9 · 2026-03-16

**Soundness:** 3
**Presentation:** 3
**Significance:** 4
**Originality:** 4
**Overall Recommendation:** 5
**Confidence:** 3

**Summary:**

The paper addresses the core shortcoming of SNNs: they're energy efficient but historically bad at generalizing to unseen categories and can't do multimodal reasoning. In response, the authors propose SVL, a pretraining framework that tries to close the gap while preserving spike efficiency.

The 3 main contributions of the paper are:
- Multi-scale Triple Alignment (MTA): a contrastive pretraining objective that aligns spike features to both CLIP's image and text spaces simultaneously, giving the SNN open-world recognition capabilities without labels
- Re-parameterizable Vision-Language Integration (Rep-VLI): converts text embeddings into classification weights offline, so the text encoder can be completely discarded at inference, keeping deployment lightweight and hardware-friendly
- Spike-driven PointFormer: the first fully spike-driven point Transformer, where self-attention reduces to sparse additions rather than expensive multiplications, enabling faster and more memory-efficient training

By employing SVL, the models trained achieve comparable or competitive results on downstream benchmarks at a substantial reduction in energy cost.

**Compliance With Llm Reviewing Policy:**

Affirmed.

**Key Questions For Authors:**

1. Can you please clarify where your training data is coming from?
2. Can you please explain more about how your 3D input representation is formed?

**Limitations:**

No - I think the authors could have talked about how building a model that can understand the 3D open-world could contribute to usage of these models in more high risk settings such as autonomous weapons.

**Strengths And Weaknesses:**

Soundness:
The technical components introduced in the paper are formulated with mathematical expressions, and the methods used are well-motivated and appropriate. The empirical experiments are extensive and well-designed covering classification, segmentation, detection, action recognition, captioning and QA. Ablations studies are also carried out across the dimension of the MTA loss combinations, timestep/firing bit configurations, discretization choices and CLIP encoder size. However, the pretraining dataset wasn't clearly specified in the paper, potentially making reproduction challenging.

Presentation:
The submission is written well, with the exception of the section about 3D input representation being hard to follow. It's unclear what P and F stand for in the collections of all points. It is also well-structured with informative figures and tables.

Significance:
The paper addresses the lack of a robust pretraining algorithm for SNNs to imbue it with strong vision/multimodal understanding in 3D open-world scenarios, where SNNs can be particularly useful due to the sparse nature of the event streams of 3D data. The paper proposes SVL and Spike-driven PointFormer, a novel framework and model architecture that is SNN-based and can generalize to downstream tasks for 3D open-world multimodal understanding. The authors also contribute Rep-VLI which removes the text encoder at inference entirely and makes deployment of SNNs compatible to neuromorphic hardware deployment. The energy and parameter efficiency claims are also substantial (e.g 204x energy advantage over ULIP-2 on Objaverse-LVIS at comparable accuracy or achieving 85.4% accuracy on ModelNet40 while consuming only 0.79mJ of energy).

Originality:
The authors contribute new methods to make SNN-based models a stronger generalizer in 3D downstream tasks. They provide new insight the lack of pretraining algorithms for SNNs and deepen understanding that it's critical to make these models efficient to deploy as well. From this, a couple of new methods are introduced (extending or combining from existing methods in the vision-language area) and they are well-motivated and well-articulated.
- MTA: extends contrastive vision-language alignment (from CLIP's 2-way to a 3-way spike/image/text objective)
- Rep-VLI: re-parameterizing offline text embeddings directly into classification weights for inference-time text-encoder-free operation
- Spike-driven PointFormer: claimed as the first fully spike-driven point Transformer; prior work (Spike Point Transformer, Wu et al. 2025) still uses non-spiking operators.

---

> ### Author Rebuttal · Authors · 2026-03-27
>
> We thank Reviewer for the thorough and positive evaluation, and for recognizing our contributions in Multi-scale Triple Alignment (MTA), Re-parameterizable Vision-Language Integration (Rep-VLI), and the Spike-driven PointFormer. We address the remaining questions below.
>
> ---
> > *Q1. Can you please clarify where your training data is coming from?*
>
> **A:** Thank you for pointing this out. We apologize for the omission and will explicitly state the pretraining data sources in the revised manuscript (Appendix D: Datasets).
>
> **Pretraining data.** Our triplet dataset $\{(3D, Image, Text)\}$ is constructed from two sources:
> - **Objaverse** [1]: We render 10 views per 3D object to obtain images and use the accompanying metadata/captions as text. Point clouds are sampled from mesh surfaces (10k points/object), yielding ~800K triplets.
> - **ShapeNet** [2]: We similarly render multi-view images and use class-level text prompts (e.g., "a 3D model of a chair"), providing ~52K additional triplets.
>
> For the SVL-13B multimodal learning experiments (Table 2), we follow the PointLLM [3] pipeline, constructing a ~660K point-text alignment dataset for Stage-1 and a ~70K instruction-following dataset for Stage-2, as detailed in Appendix C (Table 11).
>
> ---
> > *Q2. Can you please explain more about how your 3D input representation is formed?*
>
> **A:** We appreciate this question. We will revise Section 4.1 to improve clarity.
> **For point clouds:** Raw 3D data is represented as $D^t=\{\mathcal{P},\mathcal{F}\}$, where $\mathcal{P}\in\mathbb{R}^{N\times3}$ are coordinates and $\mathcal{F}\in\mathbb{R}^{N\times d}$ are per-point features (e.g., normals/colors). Points are then voxelized into regular grids (voxel backbones like E-3DSNN) or grouped by FPS+KNN (point backbones like Spike-driven PointFormer). The I-LIF neuron encodes them into spatio-temporal spike trains at timestep $t$.
>
> **For event streams (DVS):** Events are $E_i=(x_i,y_i,t_i,p_i)$. Following [4][5], we use a sliding window and normalize timestamps to a z-coordinate:
> $$z_i=\frac{t_i-t_{\min}}{t_{\max}-t_{\min}}$$
> This converts events into a spatio-temporal point cloud $E_i=(x_i,y_i,z_i)$, enabling a unified spike-driven encoder for both point clouds and event streams.
>
> **Notation clarification:** $\mathcal{P}$ denotes all 3D positions, $\mathcal{F}$ denotes per-point features, and subscript $k$ indexes individual voxels/groups.
>
> ---
> > *Q3. Regarding the limitation on ethical considerations.*
>
> **A:** We appreciate the reviewer raising this important point. We will add the following discussion on potential dual-use risks in the revised Impact Statement:
>
> "While SVL is designed to advance energy-efficient 3D perception for beneficial applications such as autonomous driving, robotics, and assistive technologies, we acknowledge that improved 3D open-world understanding could potentially be repurposed in sensitive contexts, including military applications. We advocate for responsible use and encourage the community to develop guidelines for the ethical deployment of efficient 3D perception systems."
>
> ---
> [1] Objaverse: A Universe of Annotated 3D Objects. CVPR 2023.
>
> [2] ShapeNet: An Information-Rich 3D Model Repository. Arxiv 2015.
>
> [3] PointLLM: Empowering Large Language Models to Understand Point Clouds. ECCV 2024.
>
> [4] Space-Time Event Clouds for Gesture Recognition: From RGB Cameras to Event Cameras. WACV 2019.
>
> [5] Spiking PointNet: Spiking Neural Networks for Point Clouds. NeurIPS 2024.

---

> > ### Author Rebuttal · Reviewer_jgH9 · 2026-04-04
> >
> > Thank you for the detailed responses to my questions! My questions have been acknowledged properly. I'll maintain my score.

---

> > > ### Author Response · Authors · 2026-04-06
> > >
> > > We'd be happy to add the discussion with you in rebuttal to the manuscript. We sincerely appreciate your handling of our paper, your comments helped us a lot.

---

### Decision · Program_Chairs · 2026-04-30

**Decision:**

Accept (spotlight)

**Comment:**

This paper received unanimous positive reviews: 5, 4, 4, 6. The reviewers note a wide array of strengths: the paper addresses an important problem: enabling Spiking Neural Networks (SNNs) to achieve open-world 3D understanding while preserving their energy efficiency; the paper does a good job outlining why this matters; the work is supported by extensive and well-designed experiments. While there was some confusion about method details from the initial draft, the rebuttal has provided detailed explanations that turned borderline/negative reviews into positive ones. Congratulations to the authors, this is a clear accept.